# Depletion of Nsd2-mediated histone H3K36 methylation impairs adipose tissue development and function

Lenan Zhuang [1], Younghoon Jang [1], Young-Kwon Park[1], Ji-Eun Lee [1], Shalini Jain[2], Eugene Froimchuk[1], Aaron Broun[1], Chengyu Liu[3], Oksana Gavrilova[2] & Kai Ge [1]

The epigenetic mechanisms regulating adipose tissue development and function are poorly understood. In this study, we show that depletion of histone H3K36 methylation by H3.3K36M in preadipocytes inhibits adipogenesis by increasing H3K27me3 to prevent the induction of C/EBPα and other targets of the master adipogenic transcription factor peroxisome proliferator-activated receptor-γ (PPARγ). Depleting H3K36 methyltransferase Nsd2, but not Nsd1 or Setd2, phenocopies the effects of H3.3K36M on adipogenesis and PPARγ target expression. Consistently, expression of H3.3K36M in progenitor cells impairs brown adipose tissue (BAT) and muscle development in mice. In contrast, depletion of histone H3K36 methylation by H3.3K36M in adipocytes in vivo does not affect adipose tissue weight, but leads to profound whitening of BAT and insulin resistance in white adipose tissue (WAT). These mice are resistant to high fat diet-induced WAT expansion and show severe lipodystrophy. Together, these results suggest a critical role of Nsd2-mediated H3K36 methylation in adipose tissue development and function.

---

[1] Adipocyte Biology and Gene Regulation Section, LERB, National Institute of Diabetes and Digestive and Kidney Diseases, NIH, Bethesda, MD 20892, USA. [2] Mouse Metabolism Core Laboratory, National Institute of Diabetes, Digestive and Kidney Diseases, NIH, Bethesda, MD 20892, USA. [3] Transgenic Core, National Heart, Lung, and Blood Institute, NIH, Bethesda, MD 20892, USA. These authors contributed equally: Younghoon Jang, Young-Kwon Park. Correspondence and requests for materials should be addressed to K.G. (email: kai.ge@nih.gov)

A dipose tissue plays a critical role in regulating energy balance and glucose homeostasis[1]. In mammals, there are two types of adipose tissue: white and brown. White adipose tissue (WAT) is composed of white adipocytes containing a single, large lipid droplet. WAT is mainly located in the subcutaneous and abdominal areas of the body. WAT stores excess energy in the form of triglyceride. In addition, WAT regulates energy balance through the synthesis and secretion of adipokines such as leptin, adiponectin, resistin, and angiotensinogen (Agt)[1]. Brown adipose tissue (BAT) is composed of brown adipocytes enriched with numerous mitochondria and is mainly located in the interscapular region in rodents[2]. In contrast to WAT, BAT is specialized in expending energy to generate heat. The thermogenic function of BAT is largely due to the expression of uncoupling protein-1 (Ucp1), a BAT selectively expressed mitochondrial protein[3]. In adipose tissues, the sympathetic nervous system regulates thermogenesis and lipolysis through the β3-adrenergic signaling pathway[4].

Development of adipose tissue (adipogenesis) is under the control (Ctrl) of transcriptional and epigenetic mechanisms. Adipogenic transcription factors have been studied extensively[5,6]. Peroxisome proliferator-activated receptor-γ (PPARγ), a nuclear receptor, is the master regulator in both white and brown adipogenesis[7]. Several CCAAT/enhancer-binding protein (C/EBP) family members, especially C/EBPα, activate and maintain the expression of PPARγ. C/EBPα and PPARγ promote each other's expression and cooperate to activate the expression of thousands of genes critical for the phenotypic conversion of preadipocytes to adipocytes[7]. Methylation on histone H3 lysine 4 (H3K4), H3K9, and H3K27 have also been implicated in regulating adipogenesis[8–13]. Histone methylation is also implicated in regulating adipocyte functions. For example, BAT-selective ablation of Lsd1, an H3K4 and H3K9 demethylase, leads to the down-regulation of BAT-selective genes but up-regulation of WAT-selective genes, resulting in lipid accumulation and whitening of BAT[14]. However, the role of H3K36 methylation in adipogenesis and adipocyte function has remained unclear.

H3K36 methylation has been shown to associate with active transcription. Multiple H3K36 methyltransferases have been described[15]. Setd2 is an H3K36 tri-methyltransferase responsible for the majority of H3K36me3 in cells[16]. Nsd1 and Nsd2 show specific mono-methyltransferase and di-methyltransferase activity on H3K36, generating H3K36me1 and H3K36me2[17,18]. Depletion of Nsd2 in cells decreases global H3K36me2 but does not affect H3K36me3[17,19]. Nsd2-depleted cells show increased levels of global H3K27me3[19], a finding consistent with in vitro data that H3K36 methylation antagonizes PRC2-mediated H3K27 methylation[20]. The increased H3K27me3 results in the repression of gene expression[19]. Thus, the dynamics of H3K36me2 could change the chromatin landscape of H3K27me3, thereby altering transcriptional programs.

Recently, genetic mutation of lysine 36 to methionine (K36M) of histone H3.3 was reported in chondroblastoma patients[21]. Ectopic expression of K4M, K9M, K27M, and K36M mutants of H3.3 specifically deplete endogenous H3K4, H3K9, H3K27, and H3K36 methylation, respectively[22,23]. In this study, we use K36M mutant of histone H3 as a tool to clarify the role of histone K36 methylation in adipose tissue development and function. We found that the expression of H3.3K36M in progenitor cells prevents adipogenesis in cell culture, and adipose tissue and muscle development in mice. On the other hand, expression of H3.3K36M in adipocytes has little effects on adipogenesis, but reprograms adipocyte gene expression profile and impairs adipose tissue functions. Mechanistically, H3.3K36M inhibits adipogenesis by blocking PPARγ target gene expression. Depletion

of the major H3K36 methyltransferase Nsd2, but not Nsd1 or Setd2, phenocopies the inhibitory effects of H3.3K36M on adipogenesis and PPARγ target gene expression.

## Results

**H3.3K36M increases H3K27me3 to inhibit adipogenesis.** To understand the role of histone H3K36 methylation in adipogenesis, we used retroviruses to stably express wild-type (WT) or K36M mutant of histone H3.3 in brown preadipocytes. Consistent with previous reports[22,23], ectopic expression of H3.3K36M selectively depleted global H3K36me2, mildly decreased H3K36me3 levels, and moderately increased H3K27me3 levels (Fig. 1a). Although the expression level of H3.3K36M was much lower than that of endogenous H3 (Fig. 1a), it strongly inhibited adipogenesis and associated expression of adipogenesis markers Pparg, Cebpa, and Fabp4 (Fig. 1b, c). Ectopic expression of K36M of either histone H3.1 or H3.3 also strongly inhibited adipogenesis of 3T3-L1 white preadipocytes (Supplementary Fig. 1a–c), indicating that the K36M mutant of histone H3 inhibits both white and brown adipogenesis in culture. Retroviral vector-mediated expression of H3.3K36M in C2C12 myoblasts also markedly inhibited myogenesis and the associated expression of myogenesis markers, such as Myog, Mck, and Myh (Supplementary Fig. 1d–f).

To understand how H3.3K36M inhibits adipogenesis, cells were collected before (day 0, D0), during (day 2, D2), and after (day 7, D7) adipogenesis for RNA-sequencing (RNA-Seq) analysis. In cells expressing WT histone H3.3, 1610 (9.2%) and 3348 (19.1%) genes showed over twofold up-regulation and down-regulation, respectively, from D0 to D2. Among the 1610 up-regulated genes, 380 failed to be induced at D2 in K36M-expressing cells, and therefore they are K36M-sensitive up-regulated genes (Fig. 1d). Gene ontology (GO) analysis showed that only K36M-sensitive up-regulated genes were associated with fat cell differentiation (Fig. 1e). Consistent with the GO analysis results, K36M inhibits the induction of adipocyte genes Scd1, Irs2, Lpl, and Pgc1a in addition to Cebpa (Fig. 1f).

Next, we performed chromatin immunoprecipitation followed by DNA sequencing (ChIP-Seq) to profile the chromatin landscape of H3K36me2 and H3K27me3 in WT or K36M mutant of H3.3-expressing cells before (D0) and during (D2) adipogenesis. Because traditional ChIP-Seq analysis methods did not consider the global differences between samples, we further used quantification data from Western blot (Supplementary Fig. 2a, b) as a normalization Ctrl to achieve quantitative analysis of H3K27me3 and H3K36me2. Consistent with the Western blot results in Fig. 1a and Supplementary Fig. 2a, b, ChIP-Seq data showed that H3.3K36M decreased H3K36me2 but increased H3K27me3 on gene bodies (Supplementary Fig. 2c, d). Further analysis of H3K36me2 and H3K27me3 levels on K36M-sensitive or K36M-resistant genes during adipogenesis revealed that H3.3K36M decreased the ratio of H3K36me2/H3K27me3 around the transcription start site of K36M-sensitive genes compared with H3.3WT (Supplementary Fig. 3). H3.3K36M only had mild effects on H3K36me2 and H3K27me3 levels on the Pparg gene locus by D2, which is consistent with the mild decrease of Pparg expression in K36M-expressing cells (Fig. 1g). In contrast, H3.3K36M notably decreased H3K36me2 but increased H3K27me3 levels on the gene locus encoding C/EBPα, another major regulator of adipogenesis and a direct target of PPARγ, which correlates with the impaired Cebpa induction in H3.3K36M-expressing cells (Fig. 1g). Similar results were observed on gene loci encoding other adipogenesis markers and PPARγ targets such as Scd1, Lpl, and Pgc1a (Supplementary Fig. 2e). Together, these results suggest that H3.3K36M inhibits

adipogenesis by depleting H3K36me2 and increasing H3K27me3 to prevent the induction of C/EBPα and other adipogenic genes.

**H3.3K36M inhibits PPARγ target gene expression**. Next, we investigated whether ectopic expression of adipogenic transcription factors C/EBPα or PPARγ can rescue the adipogenesis defects in H3.3K36M-expressing preadipocytes (Supplementary Fig. 4a and Fig. 2a). Ectopic expression of either C/EBPα or PPARγ failed to fully rescue the adipogenesis defect in H3.3K36M-expressing preadipocytes (Supplementary Fig. 4b, c and Fig. 2b, c). Since H3.3K36M impaired the induction of *Cebpa* and other PPARγ target genes during adipogenesis (Fig. 2c), we speculated that H3.3K36M might inhibit PPARγ target gene activation. To test this possibility, we treated sub-confluent pre-adipocytes expressing ectopic PPARγ with 0.5 μM PPARγ ligand

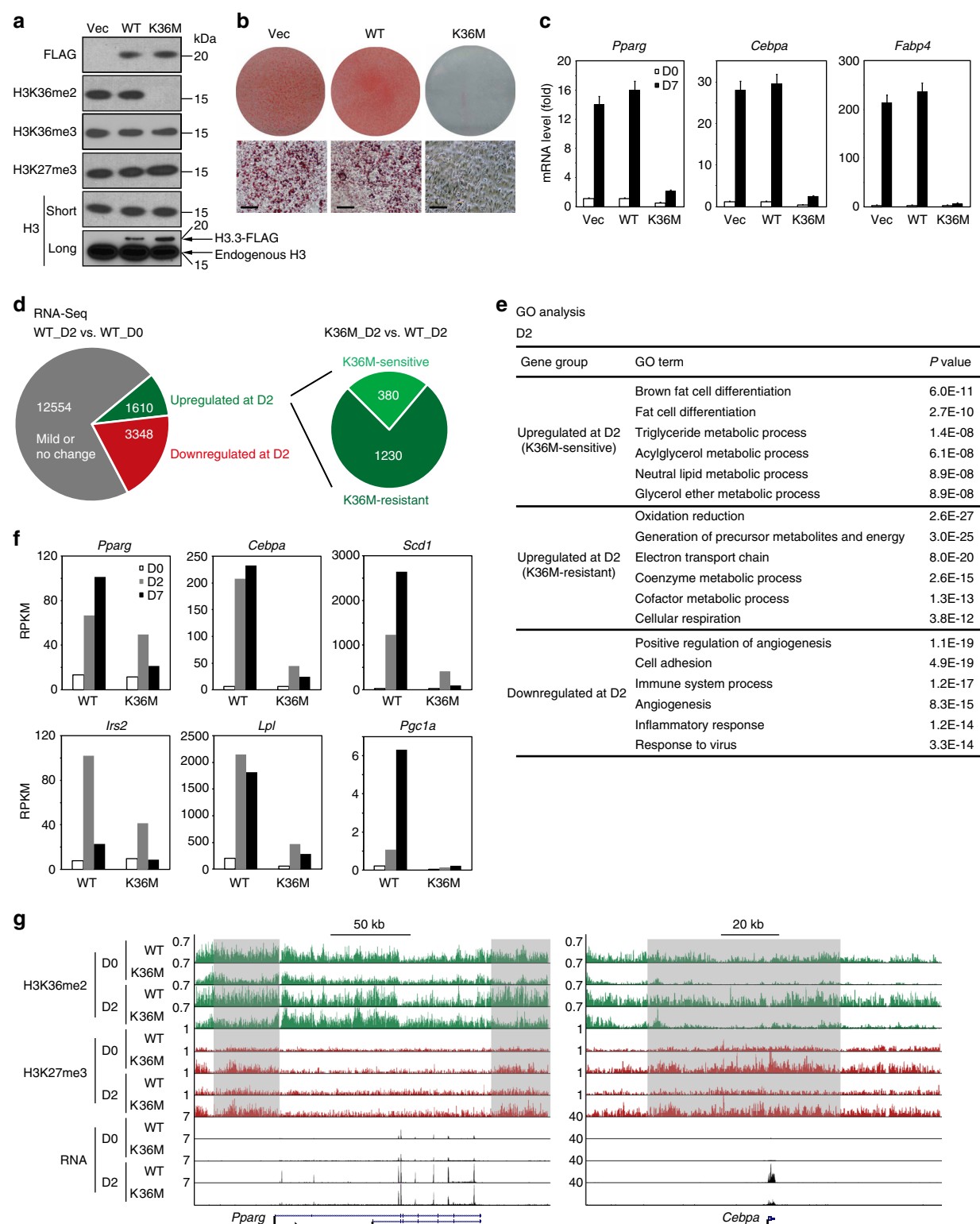

Rosiglitazone (Rosi) for 24 h, followed by RNA-Seq analysis. After 24 h Rosi treatment, 1710 (10.3%) and 925 (5.6%) genes showed over twofold up-regulation and down-regulation, respectively. Among the 1710 up-regulated genes, 708 were induced in an H3.3K36M-sensitive manner (Fig. 2d). GO analysis showed that only H3.3K36M-sensitive genes were associated with fat cell differentiation (Fig. 2e). Consistent with the GO analysis results, K36M inhibited the ligand-induced expression of PPARγ target genes *Cebpa*, *Scd1*, *Glut4*, *Irs2*, *Lpl*, *Adrb3*, and *Pgc1a* (Fig. 2f). These results indicate that H3.3K36M inhibits ligand-induced expression of PPARγ target genes critical for adipogenesis and adipocyte functions.

**H3.3K36M targets Nsd2 to inhibit adipogenesis and PPARγ.** To identify the methyltransferases important for adipogenesis that are targeted by H3.3K36M, we individually knocked down (KD) the three major H3K36 methyltransferases Nsd1, Nsd2, and Setd2 in immortalized brown preadipocytes. Depletion of Nsd1 decreased H3K36me2 but increased H3K27me3 levels. Nsd2-depleted preadipocytes showed more significant decrease of H3K36me2 and increase of H3K27me3 compared to depletion of Nsd1. Depletion of neither Nsd1 nor Nsd2 changed H3K36me3 levels (Supplementary Fig. 5a, d and Fig. 3a). In contrast, depletion of Setd2 decreased H3K36me3 but had little effect on H3K36me2 and H3K27me3 levels (Supplementary Fig. 5g). *Nsd1* or *Setd2* KD did not affect cell growth rates. Although *Nsd2* KD slightly decreased the growth rate of immortalized brown preadipocytes, these cells could grow to be over-confluent (Supplementary Fig. 6a). In adipogenesis assays, depletion of Nsd2 but not Nsd1 or Setd2 resulted in severe differentiation defects (Supplementary Fig. 5 and Fig. 3b).

By RNA-Seq analysis, we found that from D0 to D2 during adipogenesis, 1935 (11.4%) and 2980 (17.5%) genes showed over twofold up-regulation and down-regulation, respectively. Among the 1935 up-regulated genes, 507 were induced in an Nsd2-dependent manner (Fig. 3c). GO analysis showed that only the Nsd2-dependent up-regulated gene group was highly associated with fat cell differentiation (Fig. 3d). By comparing the 507 Nsd2-dependent and the 380 K36M-sensitive up-regulated genes (Fig. 1d) at D2 of adipogenesis, we found 150 genes shared by these two groups. Interestingly, *Cebpa* but not *Pparg* was among these 150 genes (Fig. 3e). Gene set enrichment analysis[24] revealed that genes down-regulated by *Nsd2* KD were significantly enriched among genes down-regulated by H3.3K36M at D2 of adipogenesis. The GO analysis of core enrichment genes revealed fat cell differentiation as the top term (Supplementary Fig. 7). These data indicate that Nsd2 depletion mimics the effect of ectopic H3.3K36M on repression of adipogenesis.

To determine whether the enzymatic activity of Nsd2 was required for adipogenesis, we used Clustered regularly interspaced short palindromic repeats (CRISPR) technology to generate *Nsd2* knockout (KO) preadipocytes (Fig. 3f). Then, we generated two mutant forms of human NSD2, Y1092A/Y1179A (double mutant 1, DM1) and H1142G/Y1179A (DM2). Y1092A, H1142G, and Y1179A mutations have been shown to eliminate the H3K36 methyltransferase activity of NSD2 separately[17,25]. The *Nsd2* KO preadipocytes were infected with retroviruses expressing either WT NSD2, DM1 or DM2. Western blot using an antibody that can detect both human and mouse Nsd2 showed that expression level of ectopic NSD2 was similar to that of endogenous Nsd2 (Fig. 3f). Deletion of endogenous *Nsd2* led to decreased H3K36me2 levels, which could be restored by expression of the WT but not the catalytic mutant forms of Nsd2 (Fig. 3f). Ectopic expression of WT NSD2, but not mutants, rescued the adipogenesis defects in *Nsd2* KO preadipocytes (Fig. 3g, h), indicating that Nsd2 enzymatic activity is required for adipogenesis.

Consistent with the ChIP-Seq data from H3.3K36M-expressing cells (Fig. 1g and Supplementary Fig. 2e), ChIP-quantitative PCR (ChIP-qPCR) revealed that H3K36me2 levels decreased and H3K27me3 levels increased on *Cebpa*, *Lpl*, and *Cd36*, but not *Pparg*, gene loci in Nsd2-depleted preadipocytes (Supplementary Fig. 8). Furthermore, ectopic expression of either C/EBPα or PPARγ failed to rescue adipogenesis defects in Nsd2-depleted preadipocytes (Supplementary Fig. 9). In addition, Nsd2-depleted preadipocytes showed severe defects in ligand-induced expression of PPARγ target genes *Cebpa*, *Cd36*, and *Lpl* (Fig. 3i). The results from Nsd2-depleted preadipocytes phenocopy those from H3.3K36M-expressing cells. Together, our data indicate that H3.3K36M targets Nsd2 to inhibit adipogenesis and PPARγ target gene expression.

**H3.3K36M impairs BAT and muscle development.** To find out whether depletion of H3K36 methylation inhibits adipogenesis in vivo, we generated a conditional H3.3K36M transgenic mouse strain, LSL-K36M. The expression of FLAG-tagged H3.3K36M driven by a CAG promoter was blocked by the insertion of four copies of SV40 stop signals (STOP) flanked by two loxP sites (Fig. 4a). We crossed LSL-K36M with *Myf5-Cre* mice to delete the STOP cassette and induce H3.3K36M expression in progenitor cells of BAT and muscle lineages[8,26]. The resulting LSL-K36M; *Myf5-Cre* E18.5 embryos showed a slightly abnormal hunched posture due to reduction of back muscles (Fig. 4b). Immunohistochemical analysis of cervical regions of E18.5 embryos revealed that expression of H3.3K36M in progenitor cells leads to reduction of BAT and muscle mass (Fig. 4c, d). Together with observations in cell culture (Fig. 1 and Supplementary Fig. 1), these data indicate that H3K36 methylation is essential for BAT and muscle development.

To confirm that the requirement of H3K36 methylation for adipogenesis and myogenesis is cell autonomous, we crossed LSL-K36M with *Cre-ER* mice to obtain primary LSL-K36M;*Cre-ER* brown preadipocytes. After immortalization, cells were treated

**Fig. 1** Expression of H3.3K36M in preadipocytes increases H3K27me3 to inhibit adipogenesis and adipogenic gene induction. **a–c** Ectopic expression of H3.3K36M in preadipocytes inhibits adipogenesis. SV40T-immortalized brown preadipocytes were infected with retroviral vector (Vec) expressing FLAG-tagged wild-type (WT) or K36M mutant histone H3.3, followed by adipogenesis assay. **a** Western blot analysis of histone modifications. Histone extracts of preadipocytes were subjected to Western blot analysis using antibodies indicated on the left. me2 and me3 refer to di-methylation and tri-methylation, respectively. Long exposure of histone H3 Western blot reveals the relative levels of ectopic H3.3 and endogenous H3. **b** Seven days after induction of differentiation, cells were stained with Oil Red O. Upper panels, stained dishes; lower panels, representative fields under microscope. Scale bars = 30 μm. **c** qRT-PCR of *Pparg*, *Cebpa*, and *Fabp4* expression at day 0 (D0) and day 7 (D7) of adipogenesis. qRT-PCR data are presented as means ± SEM. Three technical replicates from a single experiment were used. **d–f** WT and K36M cells were collected at D0, D2, and D7 for RNA-Seq analysis. **d** Schematic of identification of K36M-sensitive and K36M-resistant up-regulated genes at D2 of adipogenesis. The threshold for up-regulation or down-regulation is twofold. **e** Gene ontology (GO) analysis of gene groups defined in **d**. **f** Expression levels of representative genes are shown in RPKM (reads per kilo base of transcript per million mapped reads) values. **g** Genome browser views of ChIP-Seq and RNA-Seq data on *Pparg* and *Cebpa* loci during adipogenesis

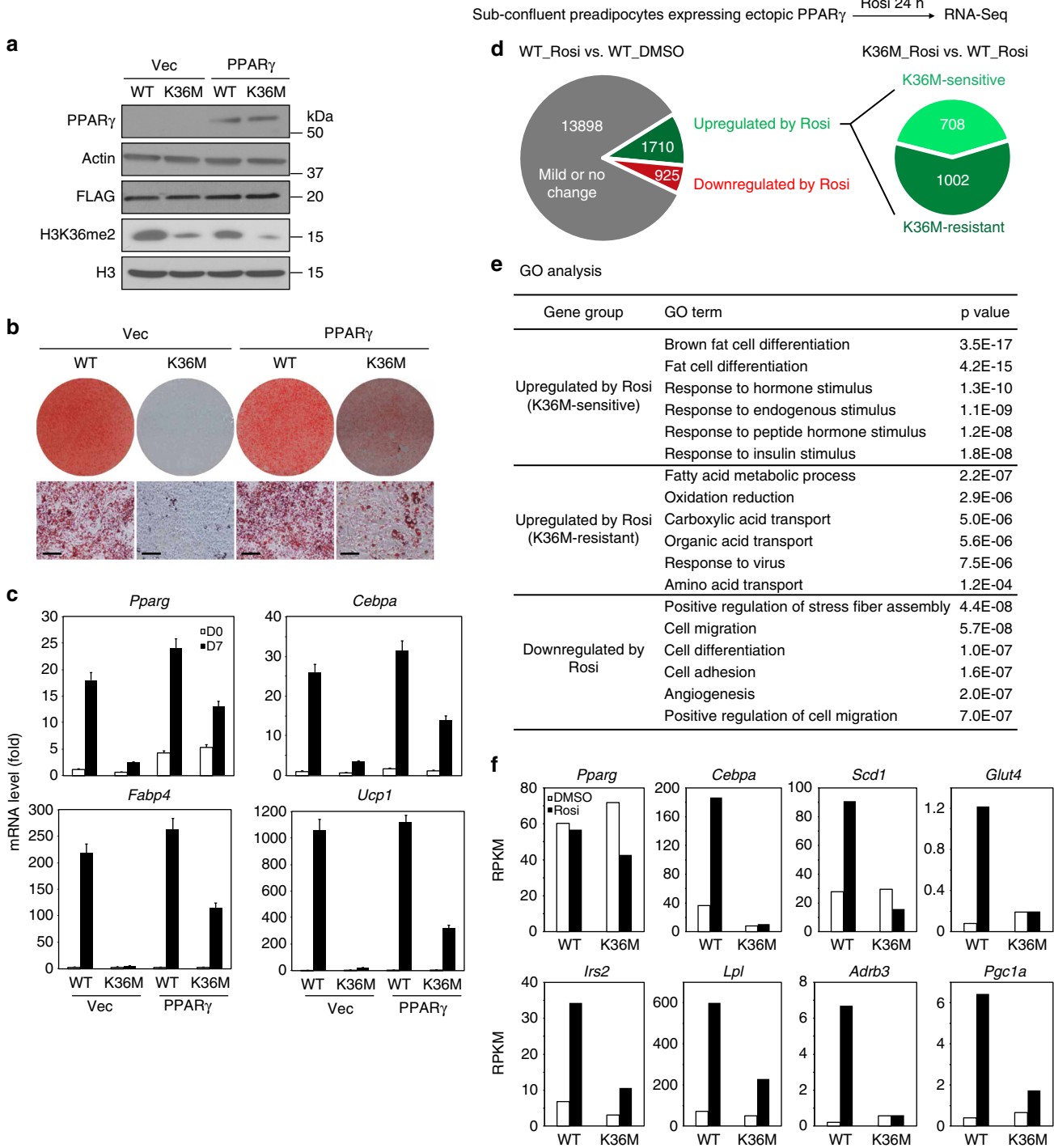

**Fig. 2** H3.3K36M inhibits ligand-induced PPARγ target gene expression. Immortalized brown preadipocytes were infected with retroviral vector expressing FLAG-tagged WT or K36M mutant histone H3.3. After puromycin selection, cells were infected with retroviral vector expressing PPARγ or empty vector, followed by hygromycin selection. **a** Western blot analysis in preadipocytes using antibodies indicated on the left. **b** Oil Red O staining at D7 of adipogenesis. Scale bars = 30 μm. **c** qRT-PCR of *Pparg*, *Cebpa*, *Fabp4*, and *Ucp1* expression at D0 and D7 of adipogenesis. qRT-PCR data are presented as means ± SEM. Three technical replicates from a single experiment were used. **d–f** Sub-confluent preadipocytes expressing ectopic PPARγ were treated with DMSO or 0.5 μM PPARγ ligand Rosiglitazone (Rosi) for 24 h, followed by RNA-Seq analysis. **d** Schematic of identification of K36M-sensitive and K36M-resistant genes up-regulated by Rosi treatment. The threshold for up-regulation or down-regulation is twofold. **e** GO analysis of gene groups defined in **d**. **f** RPKM values of *Pparg* and representative PPARγ target genes from RNA-Seq analysis

with 4-hydroxytamoxifen (4OHT) to delete the STOP cassette and induce H3.3K36M expression (Fig. 4e), followed by adipogenesis assay. Consistent with the data from LSL-K36M; *Myf5-Cre* mice, expression of H3.3K36M inhibited adipogenesis

and the induction of adipocyte marker genes such as *Pparg*, *Cebpa*, and *Fabp4* (Fig. 4f, g). For myogenesis, ectopic MyoD was expressed in the immortalized LSL-K36M;*Cre-ER* preadipocytes. 4OHT-induced H3.3K36M expression (Fig. 4h) inhibited MyoD-

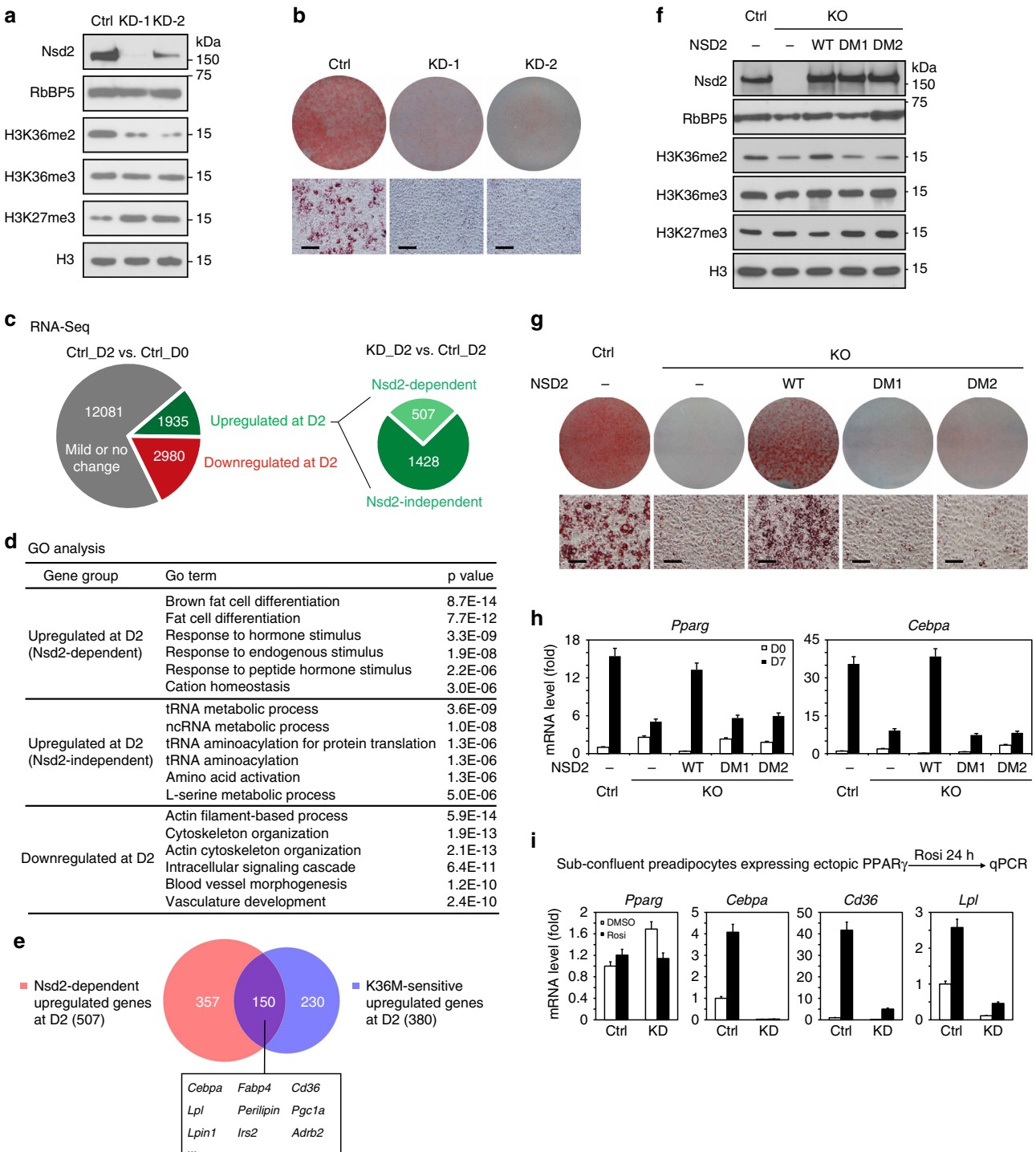

**Fig. 3** H3.3K36M targets H3K36 methyltransferase Nsd2 to inhibit adipogenesis and PPARγ target gene expression. **a–e** Immortalized brown preadipocytes were infected with lentiviral vector expressing control (Ctrl) or *Nsd2* knockdown (KD) shRNAs, followed by adipogenesis assay until D7. Cells were collected at D0 and D2 for RNA-Seq. **a** Western blot of Nsd2 and histone methylations in preadipocytes. RbBP5 and histone H3 were used as loading controls. **b** Oil Red O staining at D7 of adipogenesis. Scale bars = 30 μm. **c** Schematic identification of Nsd2-dependent and Nsd2-independent up-regulated genes at D2 of adipogenesis. The threshold for up-regulation or down-regulation is twofold. **d** GO analysis of gene groups defined in **c**. **e** Venn diagram depicting Nsd2-dependent (507) and K36M-sensitive (380) up-regulated genes at D2. **f–h** Nsd2 methyltransferase activity is required for adipogenesis. *Nsd2* knockout (KO) preadipocytes were generated using CRISPR. Control (Ctrl) and *Nsd2* KO cells were infected with retroviral vector expressing either WT, Y1092A/Y1179A mutant (DM1), or H1042G/Y1179A mutant (DM2) human NSD2, followed by adipogenesis assay. **f** Western blot of Nsd2 and histone methylations in preadipocytes. **g** Cells were stained with Oil Red O at D7 of adipogenesis. Scale bars = 30 μm. **h** qRT-PCR of *Pparg* and *Cebpa* expression at D0 and D7 of adipogenesis. **i** Nsd2 is required for ligand-induced PPARγ target gene expression. Ctrl or *Nsd2* KD preadipocytes were infected with retroviral vector expressing PPARγ. Sub-confluent cells were treated with DMSO or 0.5 μM Rosi for 24 h, followed by qRT-PCR of *Pparg* and its target genes *Cebpa*, *Cd36*, and *Lpl*. All qRT-PCR data are presented as means ± SEM. Three technical replicates from a single experiment were used

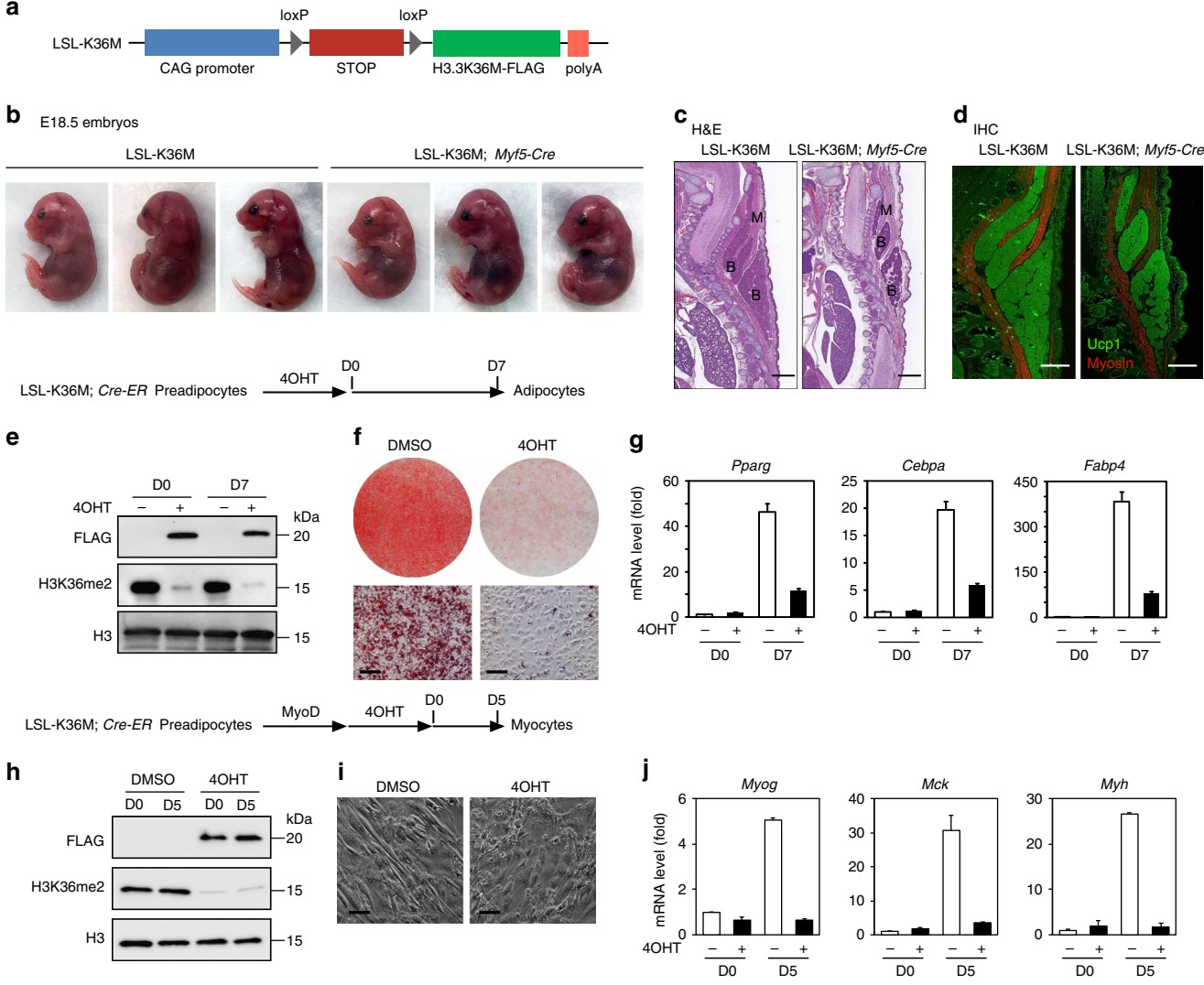

**Fig. 4** Expression of H3.3K36M in Myf5+ progenitor cells impairs brown adipose tissue and muscle development. **a** Schematic of the LSL-K36M transgenic construct. The LSL-K36M transgene consists of the following elements from 5′ to 3′: CAG promoter, four copies of SV40 stop signals (STOP) flanked by two loxP sites, H3.3K36M with FLAG tag, and polyA. **b–d** LSL-K36M were crossed with *Myf5-Cre* to generate LSL-K36M;*Myf5-Cre* mice expressing H3.3K36M in progenitor cells of brown adipose tissue (BAT) and muscle. **b** Representative morphology of E18.5 embryos. **c**, **d** Histological analysis of E18.5 embryos. Sagittal sections of cervical/thoracic area were stained with H&E (**c**), or with antibodies against the BAT (B) marker Ucp1 (green) and the muscle (M) marker Myosin (red) (**d**). Scale bar = 300 μm. **e–g** LSL-K36M were crossed with *Cre-ER* to generate LSL-K36M;*Cre-ER* mice. Primary brown preadipocytes were isolated from newborn pups. After SV40T immortalization, cells were treated with 4-hydroxytamoxifen (4OHT) to induce H3.3K36M expression, followed by adipogenesis assay. **e** Western blot in LSL-K36M;*Cre-ER* preadipocytes (D0) and adipocytes (D7). **f** Oil Red O staining at D7 of adipogenesis. Scale bars = 30 μm. **g** qRT-PCR of *Pparg*, *Cebpa*, and *Fabp4* expression at D0 and D7 of adipogenesis. **h–j** Immortalized LSL-K36M;*Cre-ER* preadipocytes were infected with retroviral vector expressing MyoD. After puromycin selection, cells were treated with 4OHT to induce H3.3K36M expression, followed by myogenesis assay. **h** Western blot in LSL-K36M;*Cre-ER* preadipocytes (day 0, D0) and myocytes (day 5, D5). **i** Cell morphologies were observed under a microscope at D5 of myogenesis. Scale bars = 20 μm. **g** qRT-PCR analysis of myogenic gene expression at D0 and D5 of myogenesis. All qRT-PCR data are presented as means ± SEM. Three technical replicates from a single experiment were used

driven myogenesis and the induction of myogenic genes such as *Myog*, *Mck*, and *Myh* (Fig. 4i, j).

**Adipose-selective expression of H3.3K36M reprograms BAT**. Next, we investigated the role of H3K36 methylation in adipose tissue functions in mice. For this purpose, we generated another transgenic mouse strain, A-K36M, in which the expression of FLAG-tagged H3.3K36M was driven by the 7.9 kb adipose-selective *Fabp4* (*aP2*) promoter described previously[27] (Fig. 5a). Different from LSL-K36M;*Myf5-Cre* mice, newborn A-K36M pups did not show significant defects in BAT development

(Supplementary Fig. 10a). While H3.3K36M was undetectable in primary A-K36M brown preadipocytes, its level increased dramatically, which caused depletion of H3K36me2 after 7 days of adipogenesis (Supplementary Fig. 10b). Consistent with the observation from newborn pups, H3.3K36M expression only mildly reduced lipid accumulation and the induction of adipocyte marker genes *Pparg* and *Cebpa* after adipogenesis (Supplementary Fig. 10c, d).

In adult A-K36M mice, H3.3K36M was detected by Western blot in interscapular BAT, inguinal WAT (ing-WAT) and epididymal WAT (epi-WAT) but not in non-adipose tissues examined (Fig. 5b). *Fabp4* promoter-driven expression of

H3.3K36M depleted endogenous H3K36me2 and H3K36me3 and mildly increased H3K27me3 in the BAT of A-K36M mice (Fig. 5c). Compared with WT Ctrl, A-K36M mice did not show significant changes in food intake, energy expenditure, activity, or $O_2$ consumption, although the respiratory exchange ratio increased slightly (Supplementary Fig. 11a–e). A-K36M mice

showed similar body weight and fat and lean mass compared to that of Ctrl mice at 8 weeks of age (Fig. 5d). The weight and size of the BAT, WATs, and other tissues were similar between A-K36M and Ctrl mice (Fig. 5e, f). These results indicate that *Fabp4* promoter-driven H3.3K36M expression has minimal effects on adipose tissue development.

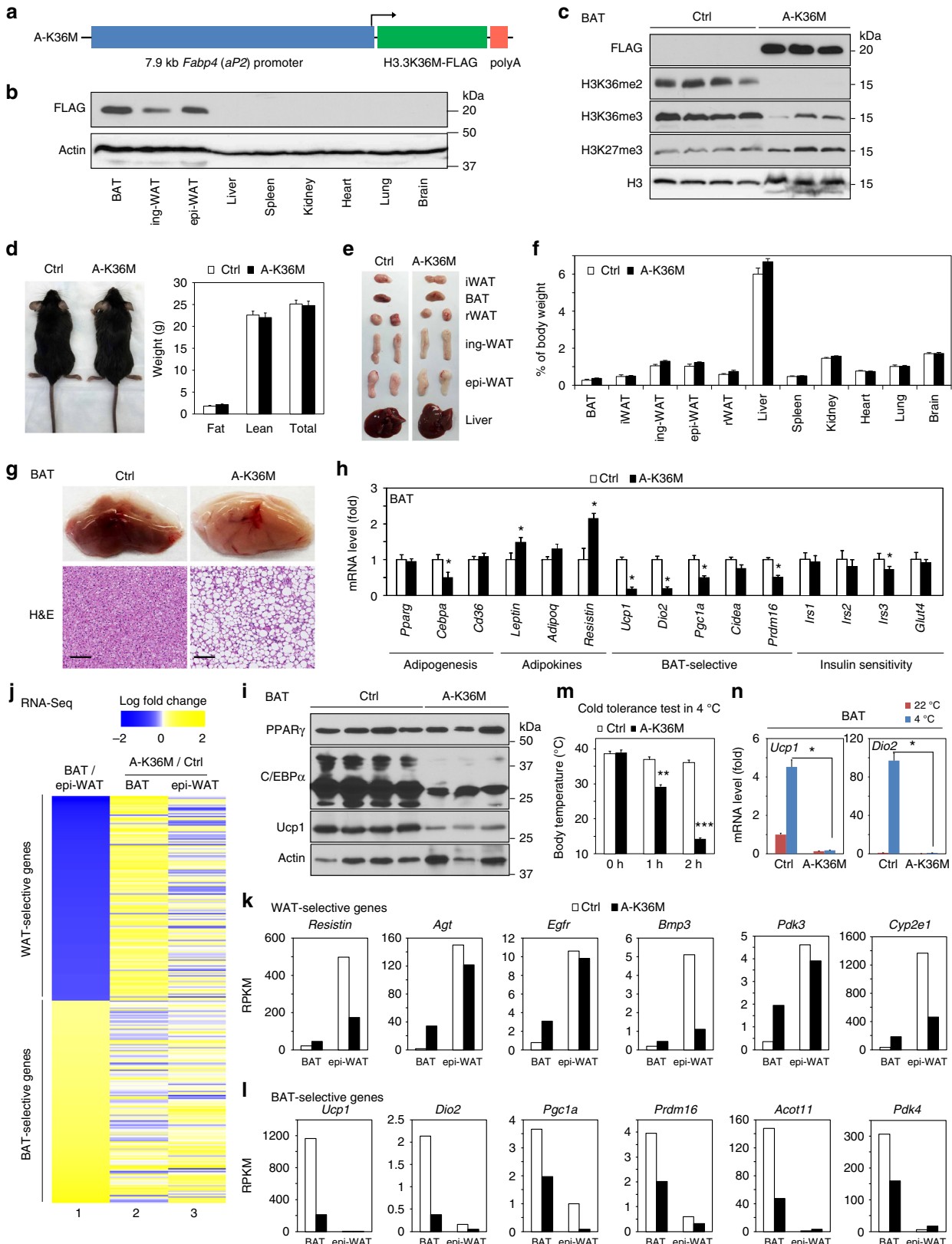

Interestingly, the BAT of A-K36M mice showed whitening in color and increased lipid accumulation (Fig. 5g). Expression levels of *Cebpa* and BAT-selective genes *Ucp1*, *Dio2*, *Pgc1a*, and *Prdm16* decreased in the BAT of A-K36M mice, while *Pparg* levels remained similar (Fig. 5h). Decreased expression of C/EBPα and Ucp1 but not PPARγ was confirmed by Western blot (Fig. 5i) and/or immunostaining (Supplementary Fig. 11f). In addition, expression of *Leptin* and *Resistin*, which are normally enriched in WAT, increased in the BAT of A-K36M mice (Fig. 5h). Consistently, H3.3K36M increased the expression of WAT-selective genes *Resistin* and *Agt* but reduced the expression of BAT-selective genes *Ucp1* and *Pgc1a* after adipogenesis of primary A-K36M brown preadipocytes (Supplementary Fig. 10d). H3K36me2 levels decreased while H3K27me3 levels increased on *Cebpa*, *Pgc1a*, and *Ucp1*, but not *Pparg* gene loci in both brown A-K36M adipocytes differentiated in culture and BAT of A-K36M mice (Supplementary Fig. 10e, f).

To further investigate the gene expression profile changes, BAT and epi-WAT were collected from A-K36M and Ctrl mice for RNA-Seq. We first determined the top 1000 WAT-selective and BAT-selective genes by comparing the transcriptomes of BAT and epi-WAT in Ctrl mice (Fig. 5j, column 1). The majority of WAT-selective genes (813) were up-regulated more than twofold in the BAT of A-K36M mice (Fig. 5j, column 2 and Fig. 5k), whereas BAT-selective genes showed reduced expression (Fig. 5l). These data suggest that depletion of H3K36 methylation by H3.3K36M causes reprogramming of the BAT gene expression profile and whitening of BAT.

A-K36M mice maintained core body temperatures when housed at room temperature (~22 °C). However, their core body temperatures dropped to ~28 °C after 1 h and to ~15 °C after 2 h exposure to an ambient temperature of 4 °C (Fig. 5m). Expression levels of thermogenesis genes *Ucp1* and *Dio2* decreased in both BAT (Fig. 5n) and ing-WAT (Supplementary Fig. 11g) of A-K36M mice at room temperature and failed to induce after 2 h cold exposure, indicating thermogenesis defects in both BAT and ing-WAT of A-K36M mice. Lipolysis is critical for thermogenesis in mice[4]. Activation of the β3-adrenergic signaling pathway by CL316,243 (CL), a selective β3-adrenergic receptor agonist, increases lipolysis and free fatty acid (FFA) release and induces energy expenditure to generate heat[28]. We found that the administration of CL increased total energy expenditure in Ctrl, but not in A-K36M mice (Supplementary Fig. 11h). A-K36M mice also showed defects in CL-stimulated lipolysis, indicated by reduced FFA release (Supplementary Fig. 11i). Together, data from A-K36M mice suggest that adipose-selective expression of H3.3K36M depletes H3K36 methylation and reprograms BAT gene expression, which has minimal effects on adipose tissue development but leads to severe thermogenesis defects in BAT.

**A-K36M mice show insulin resistance in WAT.** Next, we analyzed WAT of A-K36M mice. Hematoxylin and eosin (H&E) staining of ing-WAT and epi-WAT sections revealed similar sizes of adipocytes between A-K36M and Ctrl mice (Fig. 6a). Similar H&E staining results and triglyceride contents were observed in the liver of A-K36M and Ctrl mice (Fig. 6b). Levels of adiponectin, but not leptin, were decreased in the serum of A-K36M mice (Fig. 6c). FFA, triglyceride, and cholesterol levels in serum were similar between A-K36M and Ctrl mice (Fig. 6d). However, the expression levels of adipogenesis markers *Pparg*, *Cebpa*, and *Cd36*, lipolysis genes *Lpl*, *Hsl*, and *Adrb3*, and adipokines *Adiponectin* (*Adipoq*) and *Resistin* were decreased in both ing-WAT and epi-WAT of A-K36M mice (Fig. 6e), suggesting that depletion of H3K36 methylation by H3.3K36M impairs WAT gene expression.

In addition, insulin signaling pathway genes *Irs1* and *Irs2* were expressed at markedly lower levels in WATs (Fig. 6e), but not BAT (Fig. 5h), of A-K36M mice compared to Ctrl, suggesting impaired insulin sensitivity in A-K36M mice. Indeed, glucose tolerance test (GTT) (Fig. 6f) and insulin tolerance test (ITT) (Fig. 6g) showed that A-K36M mice were insulin resistant. Under randomly fed status or after 4 h fasting, the blood glucose levels were similar between A-K36M and Ctrl mice (Fig. 6h). However, the serum insulin levels in A-K36M mice were much higher (Fig. 6i), indicating that A-K36M mice need more insulin to maintain similar levels of blood glucose. Western blot of insulin-stimulated phosphorylation of Akt confirmed impaired insulin signaling in ing-WAT and epi-WAT, but not BAT and liver, of A-K36M mice (Fig. 6j). These data indicate that although adipose-selective expression of H3.3K36M has minimal effects on WAT development, it inhibits insulin signaling pathway gene expression in WATs, which leads to insulin resistance in mice.

**A-K36M mice are resistant to HFD-induced WAT expansion.** After analyzing A-K36M mice under regular diet (RD), we examined them under high fat diet (HFD) for 10 to 11 weeks. A-K36M mice showed significantly less body weight and fat mass, but not lean mass, than the Ctrl after HFD (Fig. 7a, b and Supplementary Fig. 12a). Ctrl and A-K36M mice had similar food intakes under RD or HFD, but A-K36M mice had higher accumulative energy expenditure compared with Ctrl mice when they were fed with HFD (Supplementary Fig. 12b, c). The ing-WAT and epi-WAT were much smaller, while the liver was much larger in A-K36M mice (Fig. 7c). Adipocytes in ing-WAT and epi-WAT of A-K36M mice were smaller, but there was more lipid accumulation in the liver (Fig. 7d). These results indicate a failure in the expansion of adipose tissues and a fatty liver phenotype in the A-K36M mice after HFD. Consistent with the markedly reduced

---

**Fig. 5** Adipose-selective expression of H3.3K36M depletes H3K36 methylation and reprograms BAT. **a** Schematic of transgenic (A-K36M) construct. Expression of FLAG-tagged H3.3K36M is under the control of the 7.9 kb adipose tissue-selective *Fabp4* promoter. **b** Tissue distribution of FLAG-tagged H3.3K36M in A-K36M mice was determined by Western blot. All data in this figure were from 8-week-old to 10-week-old male mice fed with regular diet. **c** Western blot in BAT of control (Ctrl) and A-K36M mice. **d** Representative picture of Ctrl and A-K36M mice (left). Fat mass, lean mass, and total body weight of Ctrl and A-K36M mice (*n* = 6 per group) were measured by MRI (right). **e** Representative pictures of interscapular WAT (iWAT), BAT, retroperitoneal WAT (rWAT), inguinal WAT (ing-WAT), epididymal WAT (epi-WAT), and liver. **f** Average tissue weights in Ctrl and A-K36M mice (*n* = 6 per group) are presented as % of body weight. **g** Enlarged images (upper panels) and H&E staining (lower panels) of BAT isolated from Ctrl and A-K36M mice. Scale bar = 100 μm. **h** qRT-PCR analysis in BAT of Ctrl and A-K36M mice (*n* = 6 per group). **i** Western blot of PPARγ, C/EBPα, and Ucp1 in BAT isolated from Ctrl and A-K36M mice. **j–l** BAT and epi-WAT were collected from A-K36M and Ctrl mice for RNA-Seq. **j** Heatmaps of BAT-enriched or epi-WAT-enriched genes. Column 1, log fold change between RPKM of BAT vs. epi-WAT in Ctrl mice. Top 1000 WAT-selective or top 1000 BAT-selective genes were analyzed. Columns 2–3, log fold change between RPKM of A-K36M vs. Ctrl mice (A-K36M/Ctrl) in BAT (column 2) or epi-WAT (column 3). **k, l** RPKM values of WAT-selective (**k**) or BAT-selective (**l**) genes. **m, n** Cold tolerance test. Ctrl and A-K36M mice (*n* = 6 per group) were housed at room temperature (22 °C) or in cold (4 °C) for 2 h. **m** Body temperatures. **n** qRT-PCR of *Ucp1* and *Dio2* in BAT. All values in **d**, **f**, **h** and **m**, **n** are presented as mean ± SEM. Statistical comparison between groups was performed using Student's *t* test. *$p < 0.05$, **$p < 0.01$, and ***$p < 0.005$

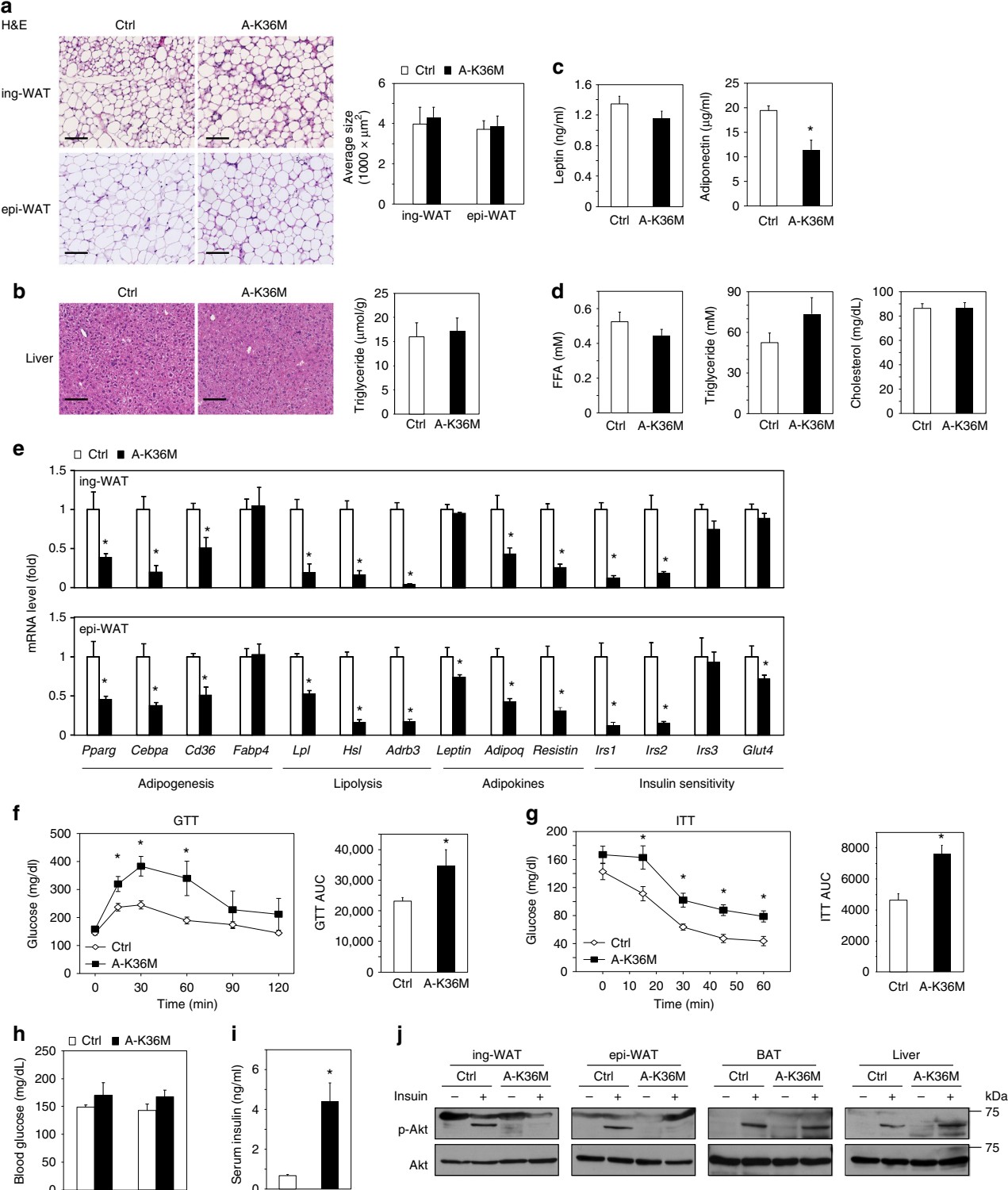

**Fig. 6** Adipose-selective expression of H3.3K36M causes insulin resistance in WAT. Eight-week-old to 10-week-old male Ctrl and A-K36M mice ($n = 6$ per group) were fed with regular diet. **a** H&E staining of ing-WAT (upper panels) and epi-WAT (lower panels). Scale bar = 100 μm. Average sizes of adipocytes in ing-WAT and epi-WAT are shown on the right. Adipocyte size was calculated using ImageJ. **b** H&E staining of the liver. Scale bar = 100 μm. Triglyceride content in the liver is shown on the right. **c** Serum leptin (left) and adiponectin (right) levels. **d** Serum levels of free fatty acid (FFA), triglyceride, and cholesterol. **e** qRT-PCR analysis of adipogenesis, lipolysis, insulin sensitivity, and adipokine genes in ing-WAT and epi-WAT. **f** Glucose tolerance test (GTT). Area under the curve (AUC) is shown on the right. **g** Insulin tolerance test (ITT). AUC is shown on the right. **h** Fed or fasted blood glucose levels. **i** Serum insulin levels in the random-fed status. **j** Western blot of phosphorylated Akt at Serine 473 (p-Akt) and total Akt in protein extracts of ing-WAT, epi-WAT, BAT, and liver. All values in Fig. 6 are presented as means ± SEM. Statistical comparison between groups was performed using Student's $t$ test. *$p < 0.05$

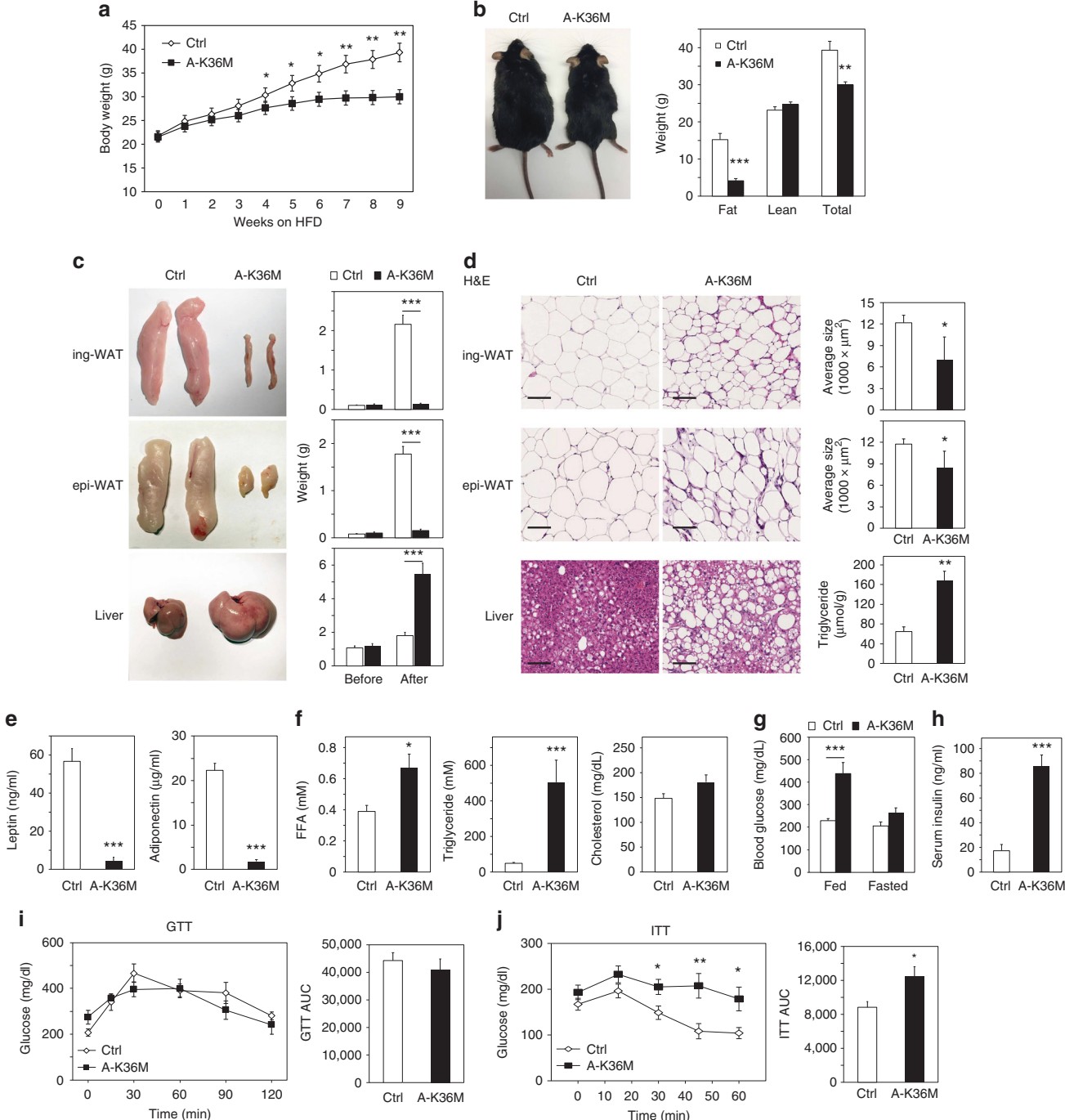

**Fig. 7** Mice with adipose-selective expression of H3.3K36M are resistant to HFD-induced WAT expansion. Male Ctrl and A-K36M mice ($n = 6$ per group) were fed with high fat diet (HFD) for 10 to 11 weeks starting from the 9th week of age. **a** Body weight during HFD feeding. **b** Representative picture of mice after HFD feeding (left). Fat mass, lean mass, and total body weight were measured by MRI (right). **c** Representative pictures of ing-WAT, epi-WAT, and liver (left panels). Average tissue weights are shown in right panels. before: tissue weights of 8-week-old mice fed with regular diet; after: tissue weights of mice after HFD. **d** H&E staining of ing-WAT, epi-WAT, and liver (left panels). Scale bar = 100 μm. Average sizes of adipocytes in ing-WAT and epi-WAT, and the liver triglyceride contents are shown on the right. **e** Serum leptin (left) and adiponectin (right) levels. **f** Serum levels of FFA, triglyceride, and cholesterol. **g** Fed or fasted blood glucose levels. **h** Serum insulin levels. **i** GTT. AUC is shown on the right. **j** ITT. AUC is shown on the right. All values are presented as mean ± SEM. Statistical comparison between groups was performed using Student's $t$ test. *$p < 0.05$, **$p < 0.01$, and ***$p < 0.005$

fat mass in the A-K36M mice, levels of adiponectin and leptin were much lower in A-K36M mice than in Ctrl (Fig. 7e). The FFA and triglyceride levels were much higher in the serum of A-K36M mice than Ctrl, but the cholesterol levels were similar (Fig. 7f). The blood glucose and serum insulin levels were much higher in

A-K36M mice after HFD (Fig. 7g, h). Although the A-K36M mice showed similar levels of glucose intolerance to Ctrl (Fig. 7i), they were more insulin resistant (Fig. 7j). The expression of genes associated with adipogenesis, lipolysis, insulin sensitivity, and adipokines decreased in WAT of A-K36M mice (Supplementary

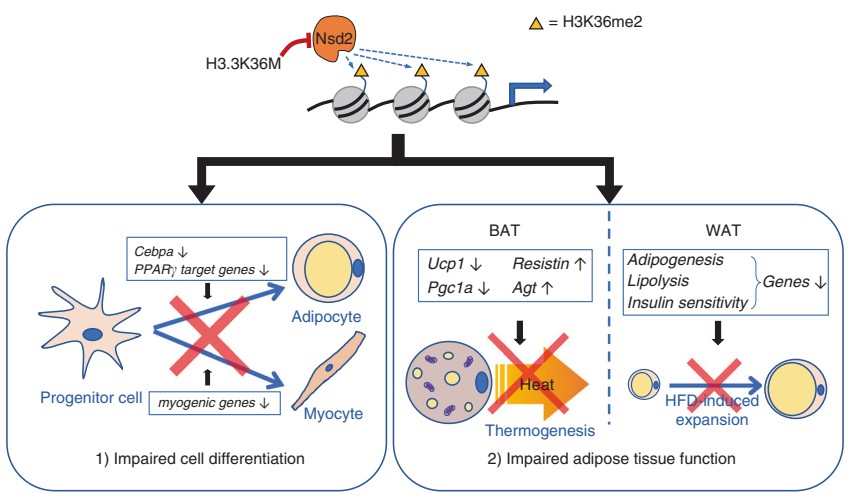

**Fig. 8** A schematic model on the role of Nsd2-mediated H3K36me2 in adipose tissue development and function. In progenitor cells, depletion of Nsd2-mediated H3K36me2 by H3.3K36M leads to defects in adipose tissue development (adipogenesis) and muscle development (myogenesis). In mature adipocytes, depletion of H3K36me2 by H3.3K36M reprograms functional gene expression, impairing brown and white adipose tissue functions

Fig. 12d). Together, these data indicate that under HFD, mice with adipose tissue-selective expression of H3.3K36M develop severe lipodystrophy associated with hyperlipidemia, insulin resistance, and diabetes. Interestingly, HFD feeding slightly decreased H3K36me2 levels in adipose tissues of Ctrl mice (Supplementary Fig. 13).

## Discussion

We show that depletion of H3K36 methylation by H3.3K36M inhibits adipogenesis by increasing H3K27me3 to prevent the induction of adipogenic gene *Cebpa* and other PPARγ target genes. H3.3K36M achieves these effects by inhibiting H3K36 di-methyltransferase Nsd2. Through generation of LSL-K36M;*Myf5-Cre* mice, we demonstrate that H3.3K36M expression in progenitor cells impairs adipose tissue development. Through generation of A-K36M mice, we show that depletion of H3K36 methylation by H3.3K36M in adipocytes reprograms gene expression profile and hinders normal functions of BAT and WAT. Together, our data suggest that Nsd2-mediated H3K36 methylation plays a critical role in regulating adipogenesis and adipose tissue function. A schematic in Fig. 8, reflecting our consistent data from in vitro and in vivo experiments, illustrates our conclusion that H3K36 methylation is important for adipose tissue development and function. In progenitor cells, depletion of H3K36 methylation by H3.3K36M leads to defects in adipose tissue development. In mature adipocytes, depletion of H3K36 methylation by H3.3K36M reprograms functional gene expression, impairing BAT and WAT functions.

A recent paper reported that H3K36M decreases H3K27me3 levels on mesenchymal stem cell (MSC) genes, which allows MSC gene expression to inhibit cell differentiation including osteo-genesis and adipogenesis[29]. Using preadipocyte differentiation as the model system, we show two mechanisms by which H3K36M inhibits adipogenesis. First, H3K36M increases H3K27me3 levels on the gene locus encoding a master adipogenic transcription factor C/EBPα, which prevents C/EBPα expression. Second, H3K36M targets Nsd2 to inhibit ligand-induced expression of target genes of PPARγ, the other master regulator of adipogen-esis. In cell culture, we show that ectopic expression of H3.3K36M inhibits adipogenesis of both white and brown preadipocytes, as well as C/EBPα-stimulated and PPARγ-stimulated adipogenesis. Furthermore, by expressing H3K36M in progenitor cells of

adipocytes, we show that H3K36 methylation is required for adipogenesis in vivo. Finally, by expressing H3K36M in mature adipocytes, we show that H3K36 methylation is important for adipose tissue function in mice.

We show in Supplementary Fig. 1a–c that H3.1K36M and H3.3K36M have similar effects on H3K36me2 levels, adipogen-esis, and adipogenic gene expression. These results are consistent with a previous study[29], which reported different distributions of H3.1K36M and H3.3K36M but similarly altered gene expression profiles between H3.1K36M-expressing and H3.3K36M-expres-sing MSCs. Together, these results suggest a substantial *trans*-effect of H3K36M, but we cannot rule out the potential invol-vement of a *cis*-effect of H3K36M. It has been shown previously that H3.3K36M incorporated in nucleosomes reduced H3K36me2 and H3K36me3 levels locally[30], suggesting a *cis*-effect. However, since the expression levels of H3.3K36M were much lower than that of endogenous H3, the *cis*-effect of H3K36M on chromatin is unlikely to play a major role in the cells used in this study.

H3.3K36M depletes H3K36me2 but only mildly decreases H3K36me3 in preadipocytes, suggesting that H3K36 di-methyltransferases may be the target of H3.3K36M. Four lines of evidence support that Nsd2 is the major target of H3.3K36M in adipogenesis. First, knockdown screening results showed that depletion of the major H3K36 di-methyltransferase Nsd2, but not the major H3K36 tri-methyltransferase Setd2, leads to defects in adipogenesis. Second, Nsd2 depletion caused defects in C/EBPα-stimulated or PPARγ-stimulated adipogenesis and impaired ligand-induced PPARγ target gene expression. Third, Nsd2 depletion caused an increase in H3K27me3, which repressed C/EBPα and other PPARγ target genes. Fourth, Nsd2 enzymatic activity is required for adipogenesis. Thus, depletion of Nsd2 phenocopies ectopic H3.3K36M expression in adipogenesis. These data indicate that H3.3K36M targets Nsd2 to deplete H3K36me2 and inhibit adipogenesis in preadipocytes. Although both Nsd1 and Nsd2 are methyltransferases for H3K36me2[15], *Nsd2* KD has more significant effects on H3K36me2 levels around gene loci of *Cebpa*, *Lpl*, and *Cd36* (Supplementary Fig. 6b), sug-gesting that Nsd2-mediated H3K36me2 plays a major role in regulating adipogenic gene expression.

Since the *Fabp4* (aP2) promoter is also active in macrophages, we have investigated the contributions of macrophages and inflammation to the phenotype of A-K36M mice. As shown in

Supplementary Fig. 14a, significant macrophage infiltration is detected in the WAT of A-K36M mice. Consistently, RNA-Seq analyses revealed increased expression of inflammatory response genes such as *Emr1* (F4/80), *Ccl2*, *Tnf*, and *IL-6* in epi-WAT of A-K36M mice (Supplementary Fig. 14b). However, RNA-Seq analyses showed that H3.3K36M mildly decreases the expression of *Emr1*, *Ccl2*, *Tnf*, and *IL-6* in RAW264.7 macrophages in cell culture (Supplementary Fig. 14c, d). Together, these data suggest that while macrophage infiltration and inflammation in WAT likely contribute to the phenotype of A-K36M mice, the inflammation is not directly caused by H3.3K36M expression in macrophages. Macrophage infiltration and inflammation in WAT have been reported in other lipodystrophic mouse models[31].

## Methods

**Plasmids and antibodies**. Retroviral plasmids expressing WT or mutants of histone H3.1 or H3.3 including pQCXIP-H3.1-FLAG, pQCXIP-H3.1K36M-FLAG, pQCXIP-H3.3-FLAG, and pQCXIP-H3.3K36M-FLAG were described[23]. pMSCVhygro-PPARγ2, pWZLhygro-C/EBPα, and pMSCVpuro-MyoD have been described[13]. The full-length human NSD2 with a C-terminal FLAG tag was from Origene (RC221350) and was cloned into pMSCVhygro to generate pMSCVhygro-NSD2-FLAG. The Y1092A/Y1179A and H1142G/Y1179A mutants of NSD2 were generated by PCR mutagenesis. Lentiviral short hairpin RNA plasmids targeting Ctrl (SHC002), mouse *Nsd1* (TRCN0000441097 and TRCN0000123379), *Nsd2* (TRCN0000253039 and TRCN0000226296), and *Setd2* (TRCN0000238537 and TRCN0000238535) genes were purchased from Sigma. For CRISPR/Cas9-mediated knockout of Nsd2, guide RNA (gRNA) was designed according to CRISPR.mit.edu. The gRNA sequence for Nsd2 is 5′-GCAATTGGTCCCCCACATAA-3′. pSpCas9 (BB)-2A-Puro (PX459) was from Addgene (48139)[32]. gRNA was cloned into PX459 to generate a gRNA-Cas9-2A-Puro single plasmid construct. All plasmids were confirmed by DNA sequencing.

Anti-NSD2 antibody (13-0002) was from EpiCypher. Anti-C/EBPα (sc-61x) and anti-PPARγ (sc-7196x) were from Santa Cruz Biotechnology. Anti-F4/80 (MAB5580) was from R&D Systems. Anti-myosin (MF20) was from Developmental Studies Hybridoma Bank. Anti-RbBP5 (AA300-109A) was from Bethyl Laboratories. Rabbit IgG (I-5006), anti-β-actin (A2228), and anti-FLAG (F3165) were from Sigma. Anti-histone H3 (ab1791) and anti-H3K36me3 (ab9050) were from Abcam. Anti-H3K36me2 (07-369) and anti-H3K27me3 (07-449) were from Millipore. Anti-Akt (9272) and anti-phosphorylated-Akt (Ser 473) (4051) were from Cell Signaling Technology. Whole-cell lysates were prepared for Western blot of Nsd2, C/EBPα, PPARγ, Ucp1, and the loading Ctrl RbBP5 and actin. Acid-extracted histones were prepared for Western blot of histone methylations. For Western blot, all antibodies were diluted to $1\ \mu g\ ml^{-1}$. Uncropped blots are available in the Supplementary Fig. 15.

**Adipogenesis and myogenesis assays**. Primary brown preadipocytes were isolated from BAT of newborn WT, LSL-K36M;*Cre-ER* or A-K36M mice. Cells were immortalized by infecting with retroviruses expressing SV40T[13]. 3T3-L1 cells were from Daniel Lane. Adipogenesis of immortalized brown preadipocytes and the 3T3-L1 white preadipocyte cell line were induced by adding induction medium (Dulbecco's modified Eagle's medium (DMEM) supplemented with 10% fetal bovine serum (FBS), 0.02 μM insulin, 1 nM T3, 0.5 mM isobutylmethylxanthine, 2 μg ml⁻¹ dexamethasone, and 0.125 mM indomethacin)[12]. After 2 days of induction, the culture medium was changed to DMEM supplemented with FBS, insulin, and T3 only. PPARγ-stimulated or C/EBPα-stimulated adipogenesis was done by infecting cells with retroviruses expressing PPARγ or C/EBPα before induction[33].

C2C12 myoblasts were purchased from ATCC and cultured in growth medium (DMEM supplemented with 15% FBS). Myogenesis was induced by changing growth medium to differentiation medium (DMEM supplemented with 2% horse serum) when cells were confluent[8].

**Quantitative reverse transcription-PCR**. Total RNA was extracted using TRIzol (Invitrogen) and reverse transcribed using ProtoScript II first-strand cDNA synthesis kit (NEB), following the manufacturer's instructions. Quantitative reverse transcription-PCR (qRT-PCR) of *Nsd2* was done using SYBR green primers: forward, 5′-GGCCAGAACAAGCTCTTACAA-3′ and reverse, 5′-TGTGGGCTCCCATAAAAGCTC-3′. Other SYBR green primers for qRT-PCR are listed in the Supplementary Table 1.

**Histology and immunofluorescence**. E18.5 embryos were isolated by Cesarean section. P0 pups were collected after birth. Samples were fixed in 4% paraformaldehyde after isolation, dehydrated in a methanol series, and embedded in paraffin. Paraffin sections were stained with routine H&E or subjected to immunohistochemistry using anti-myosin (MF20; Developmental Studies Hybridoma Bank) and anti-UCP1 (ab10983; Abcam) antibodies were diluted to 10 μg ml⁻¹[18].

**ChIP-Seq and RNA-Seq**. For ChIP-Seq analysis, crosslinking of cells was done by adding 2% formaldehyde for 10 min at room temperature, and stopped by adding 125 mM glycine. The cells were washed twice with ice-cold phosphate-buffered saline. A total of $2 \times 10^7$ cells were collected in 10 ml Farnham lysis buffer (5 mM PIPES, pH 8.0, 85 mM KCl, 0.5% NP-40, supplemented with protease inhibitors). After centrifuging at $4000 \times g$ for 5 min at 4 °C, cell pellet was washed with 10 ml Farnham lysis buffer. The nuclear pellet was resuspended in 1 ml TE buffer with protease inhibitors and sonicated for 17 min (30 s on/off cycle). Detergents were added to the lysates to make 1× RIPA buffer (10 mM Tris-Cl, pH 7.7, 1 mM EDTA, 0.1% sodium dodecyl sulfate (SDS), 0.1% sodium deoxycholate (Na-DOC), 1% Triton X-100) and centrifuged at $13,000 \times g$ for 15 min at 4 °C to remove debris. Eight micrograms of antibodies were pre-incubated with 50 μl Dynabeads Protein A (Life Technologies) overnight at 4 °C. For each ChIP, antibody–beads complex was added to chromatin and incubated overnight at 4 °C. The beads were washed with RIPA buffer twice, high-salt-RIPA buffer (RIPA containing 0.3 M NaCl) twice, LiCl buffer (50 mM Tris-Cl, pH 7.5, 250 mM LiCl, 0.5% NP-40, 0.5% Na-DOC) twice and TE buffer once. DNA was eluted and reverse-crosslinked in 200 μl elution buffer (1% SDS, 0.1 M NaHCO₃, and supplemented with 20 μg proteinase K) overnight at 65 °C, and purified by QIAquick PCR Purification Kit (Qiagen)[8,34]. ChIP SYBR Green primers were designed at indicated distance to the transcription start site of *Cebpa*. The sequences of primers are listed in Supplementary Table 1.

For RNA-Seq, mRNAs were purified using Dynabeads mRNA Purification Kit (Invitrogen), and then double-stranded cDNAs were synthesized by SuperScript Double-Stranded cDNA Synthesis Kit (Invitrogen). All ChIP-Seq and RNA-Seq sequencing libraries were constructed using NEBNext Ultra II DNA Library Prep Kit for Illumina (NEB), following the manufacturer's instructions. All ChIP-Seq and RNA-Seq samples were sequenced on the Illumina HiSeq 2500[35].

ChIP-Seq data were analyzed by SICER[36]. Raw reads were mapped to the mouse genome (mm9). H3K36me2 or H3K27me3 signal intensity at each nucleotide was calculated as read coverage, followed by scaling normalization to ensure that the average intensity across the whole genome equals to 1 for each sample. The input intensity was subtracted from the ChIP signal based on a Poisson model[17]. Then, H3K36me2 or H3K27me3 signal intensity was further normalized with global H3K36me2 or H3K27me3 levels measured by Western blot to justify the global changes of these histone methylations. ChIP-Seq and RNA-Seq datasets were deposited in the GEO database (accession no. GSE83793).

**Generation of transgenic mice**. To generate LSL-K36M transgenic mice, full-length H3.3 with a K36M point mutation and a C-terminal FLAG tag was fused downstream of CAG promoter with a loxP-STOP-loxP cassette in the middle as described previously[37]. To generate A-K36M transgenic mice, FLAG-tagged H3.3K36M was fused downstream of the 7.9 kb *Fabp4* (*aP2*) promoter described previously[27]. The transgenic DNA fragment was gel purified and injected into zygotes harvested from C57BL/6J mice. Founder mice were identified by genotyping and Western blot using an anti-FLAG antibody. LSL-K36M mice were crossed with *Myf5-Cre* (Jackson no. 007893, C57BL/6J and 129S4/SvJaeSor mixed background) or *Cre-ER* (Jackson no. 008463) to generate LSL-K36M;*Myf5-Cre* or LSL-K36M;*Cre-ER* mice. Animals were maintained on a 12 h light/dark cycle (6:00 a.m./6:00 p.m.) and standard pellet diet (NIH-07, 15 kcal% fat, Envigo Inc.), unless otherwise indicated. HFD D12492 (Research Diet) consisted of 59.4 kcal% fat, 16.2 kcal% protein, and 24.5 kcal% carbohydrates. Animals were not randomized and the researchers were not blinded during the experiment and when assessing the outcome. No animals were excluded from the analysis. All mouse experiments were performed in accordance with the NIH Guide for the Care and Use of Laboratory Animals and approved by the Animal Care and Use Committee of NIDDK, NIH.

**Body composition and metabolic studies**. Body composition was measured with the EchoMRI 3-in-1 analyzer (Echo Medical Systems). Food intake, O₂ consumption, and CO₂ production were measured at 22 °C over a 24 h period in a CLAMS system (Columbus Instruments Inc.; 2.5 L chambers with plastic floors, using 0.6 L min⁻¹ flow rate, one mouse per chamber) after a 48 h adaptation period. Motor activity (total and ambulatory) was simultaneously measured by infrared beam interruption. Resting O₂ consumption was calculated as the mean of the points with fewer than six ambulating beam breaks per minute. CL316,243 (Sigma-Aldrich; 0.01 mg kg⁻¹ intraperitoneally (i.p.)) or saline vehicle was administered and O₂ consumption was measured at 30 °C from 1 to 4 h after injection. To measure the cumulative body composition, food intake, and energy expenditure, mice were single-housed and allowed to acclimate for 1 week before measurement during the subsequent 8 weeks (3 weeks fed with RD, followed by 5 weeks fed with HFD). Energy expenditure was calculated by energy balance technique[38].

**Cold tolerance test**. To test mouse cold tolerance, mice were individually caged and exposed to an ambient temperature of 4 °C. Core body temperature was measured using a rectal thermometer (TH-5, Braintree Scientific) before and hourly after cold exposure.

**GTTs and ITTs**. For GTTs, 4 h fasted mice received glucose (1 g kg⁻¹ i.p.). For ITTs, mice were fasted for 4 h before administration of insulin (Humulin, 0.75 mI

U g$^{-1}$, i.p., Eli Lilly). For both tests, glucose level was measured from the tail vein at indicated time points with a glucometer (Contour, Bayer).

**In vivo lipolysis**. Randomly fed mice were injected i.p. saline or CL316,243 (Sigma, 0.1 mg kg$^{-1}$) in a cross-over manner and blood was collected 20 min later. Plasma FFA level was measured with reagents from Roche Diagnostics.

**Serum and tissue chemistries**. Glucose was measured with a glucometer (Bayer). FFA, triglyceride, and total cholesterol were measured with reagents from Roche Diagnostics GmbH, Pointe Scientific Inc., and Thermo Scientific, respectively. Adiponectin, insulin, and leptin were measured by enzyme-linked immunosorbent assay from ALPCO, CrystalChem, and RD Systems, respectively. Liver triglyceride was extracted using phenol–chloroform (3:1) and measured by a kit from Pointe Scientific Inc.

**In vivo insulin signaling assay**. In vivo insulin signaling in the adipose tissues and liver was measured by AKT phosphorylation[39]. Briefly, mice were fasted for 24 h before insulin injection. Anesthetized mice were surgically operated, and one side of the BAT, ing-WAT, and epi-WAT and a piece of the liver were excised and snap-frozen in liquid nitrogen for use as an untreated Ctrl. Three minutes after injection via the inferior vena cava with 1 U kg$^{-1}$ of human insulin (Eli Lilly), the other side of the BAT, ing-WAT, and epi-WAT and a piece of the liver were snap-frozen for subsequent protein extraction and Western blot analysis.

**Statistics**. Results are expressed as means ± SEM (standard error of the mean). Means of continuous outcome variables were tested with two-tailed Student's *t* test. P values <0.05 were considered statistically significant.

**Data availability**. All datasets described in the paper have been deposited in NCBI Gene Expression Omnibus under accession number GSE83793. We declare that the data supporting the findings of this study are available within the article and its Supplementary Information files, or available from the authors upon request.

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

## Acknowledgements

We thank Zhiguo Zhang for retroviral plasmids expressing WT or mutant histone H3.1 or H3.3, Binbin Lai and Weiqun Peng for help with bioinformatics analysis, Hui Sun and Min Chen for in vivo insulin signaling assay, Yinyan Ma for the serum and tissue chemistries experiments, Victoria Noe-Kim for proofreading, and Chaochen Wang for scientific discussions and NIDDK Genomics Core for sequencing. This work was supported by the Intramural Research Program of the NIDDK, NIH to K.G.

## Author contributions

Experiments were designed by L.Z., O.G., and K.G. Experiments were performed by L.Z., Y.J., Y.-K.P., J.-E.L., S.J., E.F., A.B., C.L., and O.G. Data were interpreted by L.Z., O.G., and K.G. The paper was written by L.Z. and K.G.

## Additional information

**Competing interests:** The authors declare no competing interests.

