## [Peer Review File · Nature Communications]

Reviewers' comments:

Reviewer #1 (Remarks to the Author):

The study by Zhuang et al. reports the role of H3.3K36 in adipocyte development. The authors found that ectopic overexpression of H3.3K36 mutant (K to M) increases H3K27me3 and inhibits adipogenesis. Depletion of Nsd2, a H3K36 methyltransferase, similarly inhibits adipogenesis by repressing CEBPalpha, PPARg and their target genes in cultured adipocytes. The authors further examined the role of the H3.3 mutant in vivo by characterizing the metabolic phenotype of mice expressing H3.3K36M by the Ap2 promoter. The transgenic mice displayed a "whitening" of BAT, yet resistant to high-fat diet induced body weight gain. The authors provide large amounts of data to demonstrate the role of H3.3K36 in cultured cells and in vivo. The data are intriguing and provide new insights into the epigenetic regulation of adipose tissue development and function. On the other hand, this paper suffers from several major points and needs further clarifications:

Major comments:

1. In cultured adipocytes, H3.3K36M expression near completely blocks adipogenesis, whereas the adipose tissue in the transgenic mice possesses many differentiated adipocytes under a regular diet. Reduced adipose tissue size was seen only under a high-fat diet. Furthermore, the BAT from transgenic mice expressed higher levels of white fat marker genes, whereas this phenotype was not observed in cultured cells (impaired adipogenesis in cultured cells). These results indicate a discrepancy between cultured cells and animal studies, but this aspect remains completely unclear.
2. It is unclear why energy expenditure of the ap2-H3.3K36M transgenic mice are higher than the control mice under a high-fat diet, even though BAT displays a whitening phenotype and BAT thermogenesis is impaired in TG mice.
3. Relating to the above comment, the data under a high-fat diet study suggest diet-dependent modifications in chromatin regulation through H3.3K36, but this aspect is neglected in this study.
4. It has been known the ap2-enhancer drives the expression of a transgene in macrophages and the central nervous system. This raises a concern when the phenotype is linked to inflammation.

Specific comments:

1. In RNA-Seq analysis in Fig.1, the authors need to clarify the group comparison – is the comparison between vector vs. H3.3 or between WT vs. mutant H3.3K36?
2. Regarding the mouse metabolic phenotyping, the authors should examine whole-body energy expenditure, food intake, and movement/activity of the mice under a regular diet as well. In addition, BAT thermogenesis capacity should be determined by measuring oxygen consumption rate, in addition to gene expression.
3. In Page6 line 111-112: the meaning of the sentence "traditional CHIP-seq analysis methods did not consider the global changes between samples" is unclear. Many studies have compared the data between samples. The normalization methods in Fig.2A-B and the data without "normalization" should be provided and discuss the rationale.

Reviewer #2 (Remarks to the Author):

Depletion of Nsd2-mediated histone H3K36 methylation reprograms adipose tissue development and function – zhuang et al

This manuscript identifies a role for NSD2 and its catalytic product H3K36me2 in adipogenesis in vitro and for H3K36 methylation in vivo in WAT and BAT development and function. The authors show that in immortalized cell lines the expression of H3.3K36M or depletion of NSD2 (but not NSD1 or SETD2), both of which lead to loss of H3K36me2 and a concomitant increase in H3K27me3, block differentiation of immortalized preadipocytes into adipocytes. Genomic studies (RNA-seq and ChIP-seq) are performed in both K36M and NSD2-KD-expressing cells and support a model in which loss of H3K36me2 leads to re-distribution of H3K27me3 and repression of key developmental target genes. In vivo, the authors observe a compelling phenotype in mice expressing H3.3 K36M in adipocytes. While the observed phenotype is not entirely consistent with the one observed in their tissue culture model (there does not appear to be a similar block in differentiation of pre-adipocytes), it is apparent that adipose tissue function is perturbed in the K36M mice, and there are changes in K36me2 and K27me3 that are consistent with the cell culture observations.

In summary, this manuscript identifies H3K36me2 (and thereby NSD2) crucial for normal fat development. The conclusion that this abundant histone modification is important for a developmental process is not surprising and the proposed mechanism (via H3K27me3 redistribution) has been previously described. Nonetheless, the thoroughness of the study, the in vivo mouse model work, and establishing the link between NSD2-H3K36me2 and proper physiologic fat development is significant and the work should be of broad interest to the field.

Comments:

- 1) Figures 1 & 2: While the results appear similar, no comparisons are made between changes in gene expression and K36/K27 methylation. The data should be analyzed more extensively. For example, in Fig. 1D, a distinction is made between "K36M-sensitive" and "K36M-resistant" DEGs upon differentiation. The authors seem to suggest that these changes in the K36M category are due to changes in K36/K27 methylation, but this is only explored at 4 hand-selected genes in Fig 2E. To make better use of the data they already have, the authors should generate metaplots similar to Fig 2C and 2D, but instead of "All genes", looking at "K36M-resistant genes" and "K36M-sensitive genes". The authors' model would predict that changes in K36/K27 might be observed in the latter and not the former.
- 2) Can the authors comment on the distribution of H3.3 K36M? Is it enriched at the affected loci, directly inhibiting K36me2 in cis, or are the observed effects predominantly a trans-effect? ChIP on the FLAG-tagged H3.3 at the four loci in Fig. 2E would be informative.
- 3) Figure 4: Similar to Fig. 1 & 2, the information presented in Fig. 4 appears to be consistent with results observed in K36M cells, but the direct comparison is never made. Specifically, Fig. 4D is intentionally displayed in a similar format to Fig. 1D; but do the categories defined in each figure overlap? A Venn diagram comparing the categories defined in Fig. 1D and Fig. 4D should be included in Figure 4.
- 4) Knock-down of NSD2 is known to be toxic to many cell types and can cause loss of proliferation. Thus, are the effects on adipogenesis a direct result of the loss of H3K36me2 due to expression of H3.3K36M and NSD2 KD or is it a secondary effect of toxicity associated with depletion of H3K36me2? To rule out the possibility of toxicity, the authors should determine growth rates of cells expressing H3.3K36M and control as well as Control, shNSD1, shNSD2, and shSETD2 cells without differentiation.
- 5) It is surprising that NSD1 KD led to a detectable loss of H3K36me2 (Fig S3) as the majority of H3K36me2 being generated in these cells appears to be due to NSD2 (e.g. Fig 4). Please comment.

Response to Reviewers' comments:

Summary of changes in the revised version:

We are very grateful for the highly constructive and insightful comments by the two reviewers on our previous submission, which has greatly helped us improve this work. In the revised manuscript entitled "**Depletion of Nsd2-mediated H3K36 methylation impairs adipose tissue development and function**", we have included extensive new experimental data and computational analyses to strengthen the conclusions. We believe we have fully addressed all comments, including both reviewers' concerns on the disconnect between *in vitro* and *in vivo* data, as well as the reviewer #2's concern on novelties of our study. The major changes are briefly outlined below:

1. To address both reviewers' concerns on the disconnect between *in vitro* and *in vivo* data, we have done the following:

a) Regarding the role of H3K36 methylation in adipose tissue development (adipogenesis), our previous manuscript only provided *in vitro* data to demonstrate that H3K36M inhibits adipogenesis. In the revised version, we now show *in vivo* data from a new transgenic mouse model that H3K36M inhibits adipose tissue development (**new Figure 4**).

b) Regarding the role of H3K36 methylation in adipose tissue function, our previous manuscript showed that *aP2* (*Fabp4*) promoter-driven H3K36M expression in adipocytes has little effects on the size and weight of BAT and WAT but reprograms gene expression and causes functional defects in mouse adipose tissues *in vivo*. To provide mechanistic data to explain the mouse phenotype, we now show *in vitro* data demonstrating that *aP2* promoter-driven H3K36M expression in adipocytes decreases H3K36me2 while increases H3K27me3 levels and reprograms the expression of adipocyte-specific functional genes (**new supplementary Fig. 10**)

2. Regarding the reviewer 2's concern on novelties, we have updated the manuscript to include new data and to explicitly state the novelties of our study:

a) By expressing H3K36M in progenitor cells of adipocytes, we show for the first time that H3K36 methylation is required for adipogenesis *in vivo* (**new Figure 4**).

b) A recent Science paper reported that H3K36M **decreases** H3K27me3 levels on mesenchymal stem cell (MSC) genes, which allows MSC gene expression to inhibit cell differentiation including osteogenesis and adipogenesis¹. Using preadipocyte differentiation as the model system, we show two novel mechanisms by which H3K36M inhibits adipogenesis. First, H3K36M **increases** H3K27me3 levels on the gene locus encoding a master adipogenic transcription factor C/EBP α , which prevents C/EBP α expression (**Figure 1g**). Second, H3K36M targets Nsd2 to inhibit ligand-induced expression of target genes of PPAR γ , the other master regulator of adipogenesis (**Figure 2**).

c) By expressing H3K36M in adipocytes, we show for the first time that H3K36 methylation is important for adipose tissue function *in vivo* (**Figure 5 – Figure 7**).

3. We now include a schematic to illustrate our findings (**new Figure 8**).

Reflecting our consistent *in vitro* and *in vivo* data, this schematic illustrates our conclusion that H3K36 methylation is important for both development and function of adipose tissues. In progenitor cells, depletion of H3K36 methylation by H3K36M leads to defects in adipose tissue development (adipogenesis). In mature adipocytes, depletion of H3K36 methylation by H3K36M reprograms functional gene expression, impairing brown and white adipose tissue functions.

4. To address reviewers' specific comments, we have also included **new Supplementary Fig. 3, 6, 7, 10, 13 and 14**.

Point-by-point responses:

Reviewer #1 (Remarks to the Author):

“The study by Zhuang et al. reports the role of H3.3K36 in adipocyte development. The authors found that ectopic overexpression of H3.3K36 mutant (K to M) increases H3K27me3 and inhibits adipogenesis. Depletion of Nsd2, a H3K36 methyltransferase, similarly inhibits adipogenesis by repressing CEBPalpha, PPARγ and their target genes in cultured adipocytes. The authors further examined the role of the H3.3 mutant *in vivo* by characterizing the metabolic phenotype of mice expressing H3.3K36M by the Ap2 promoter. The transgenic mice displayed a “whitening” of BAT, yet resistant to high-fat diet induced body weight gain. The authors provide large amounts of data to demonstrate the role of H3.3K36 in cultured cells and *in vivo*. The data are intriguing and provide new insights into the epigenetic regulation of adipose tissue development and function. On the other hand, this paper suffers from several major points and needs further clarifications:

Major comments:

1. In cultured adipocytes, H3.3K36M expression near completely blocks adipogenesis, whereas the adipose tissue in the transgenic mice possesses many differentiated adipocytes under a regular diet. Reduced adipose tissue size was seen only under a high-fat diet. Furthermore, the BAT from transgenic mice expressed higher levels of white fat marker genes, whereas this phenotype was not observed in cultured cells (impaired adipogenesis in cultured cells). These results indicate a discrepancy between cultured cells and animal studies, but this aspect remains completely unclear.”

RE: We thank Reviewer #1’s careful reading of our manuscript and appreciate his/her highly constructive comments.

Our study aims to understand the role of histone H3K36 methylation in both development (adipogenesis) and function of adipose tissue. Since *Fabp4* (*aP2*) promoter-driven H3.3K36M expression is relatively late in adipogenesis^{2,3}, *aP2*-H3.3K36M transgenic mice (A-K36M) are suitable for studying the role of H3K36 in adipose tissue function but not adipogenesis. Indeed, *aP2* promoter-driven H3.3K36M expression has minimal effects on adipose tissue development (adipogenesis). Newborn A-K36M pups did not show significant defects in BAT development (**new Supplementary Fig 10a**). Adult A-K36M mice showed similar body weight and fat and lean mass, and similar weight and size of the BAT and WATs compared to those of Ctrl mice (**Figure 5d-f**).

To support the critical role of H3K36 methylation in adipogenesis *in vivo*, we have added data from a new transgenic mouse model (LSL-K36M, see the **new Figure 4**). We crossed LSL-K36M mice with *Myf5-Cre* mice to induce H3.3K36M expression in progenitor cells of BAT and muscle lineages^{4,5}. Consistent with *in vitro* adipogenesis data (**Figure 1**), expression of H3.3K36M in *Myf5*⁺ progenitor cells *in vivo* led to significant defects in BAT development (**new Figure 4**).

Regarding the role of H3K36 methylation in adipose tissue function, we previously showed that *aP2* promoter-driven H3.3K36M expression in adipocytes (A-K36M mice) reprograms gene expression and causes functional defects in adipose tissues (**Figure 5-7**). In the revised version, we show *in vitro* data consistent with the functional defects *in vivo*. In primary A-K36M brown adipocytes, *aP2* promoter-driven H3.3K36M expression changed H3K36/K27 methylation levels and increased white fat marker gene expression (**new Supplementary Fig 10**).

To help understanding of our findings, we have included a schematic in the **new Figure 8**. Reflecting our consistent data from *in vitro* and *in vivo* experiments, this model illustrates our conclusion that H3K36 methylation is important for adipose tissue development and function. In progenitor cells, depletion of H3K36 methylation by H3.3K36M leads to defects in adipose tissue development

(adipogenesis). In mature adipocytes, depletion of H3K36 methylation by H3K36M reprograms functional gene expression, impairing brown and white adipose tissue functions.

“2. It is unclear why energy expenditure of the *aP2*-H3.3K36M transgenic mice are higher than the control mice under a high-fat diet, even though BAT displays a whitening phenotype and BAT thermogenesis is impaired in TG mice.”

RE: The *aP2*-H3.3K36M transgenic mice (A-K36M) show severe lipodystrophy and slightly higher cumulative energy expenditure after high-fat diet (HFD) (**Supplementary Fig 12c**). The higher energy expenditure phenotype has been observed in other lipodystrophic mice⁶ or human patients⁷. Why A-K36M mice under HFD as well as other lipodystrophic mice show increased energy expenditure will require further investigation. We believe this is beyond the scope of this manuscript.

“3. Relating to the above comment, the data under a high-fat diet study suggest diet-dependent modifications in chromatin regulation through H3.3K36, but this aspect is neglected in this study.”

RE: Following the reviewer’s suggestion, we performed Western blot of H3K36me2 in mouse BAT and epi-WAT under regular diet (RD) and HFD (**new Supplementary Fig 13**). HFD feeding slightly but reproducibly decreased H3K36me2 levels in adipose tissues. This result suggests a potential role of H3K36me2 in protection against HFD-induced obesity. Future work will be needed to find out the role of H3K36M depleted H3K36 methylation in HFD-induced obesity.

“4. It has been known the *aP2*-enhancer drives the expression of a transgene in macrophages and the central nervous system. This raises a concern when the phenotype is linked to inflammation.”

RE: First, our Western blot analyses using whole brain lysates showed that H3.3K36M protein is undetectable in the brain of *aP2*-K36M transgenic (A-K36M) mice (**Figure 5b**). Second, we have investigated the contributions of macrophages and inflammation to the phenotype of A-K36M mice. As shown in the **new Supplementary Fig 14a**, significant macrophage infiltration is detected in the WAT of A-K36M mice. Consistently, RNA-Seq analyses revealed increased expression of inflammatory response genes such as *Emr1* (F4/80), *Ccl2*, *Tnf* and *IL-6* in epi-WAT of A-K36M mice (**new Supplementary Fig 14b**). However, RNA-Seq analyses showed that H3.3K36M mildly decreases the expression of *Emr1*, *Ccl2*, *Tnf* and *IL-6* in RAW264.7 macrophages in cell culture (**new Supplementary Fig 14c-d**). Together, these data suggest that while macrophage infiltration and inflammation in WAT likely contribute to the phenotype of A-K36M mice, the inflammation is not directly caused by H3.3K36M expression in macrophages. Macrophage infiltration and inflammation in WAT have been reported in other lipodystrophic mouse models⁸.

“Specific comments:

1. In RNA-Seq analysis in Fig.1, the authors need to clarify the group comparison – is the comparison between vector vs. H3.3 or between WT vs. mutant H3.3K36?”

RE: In **Figure 1d** the comparisons are between D2 and D0 in H3.3WT-expressing cells (left pie chart) and between K36M- and WT-expressing cells at D2 (right pie chart). We have added labels to clarify the comparisons between groups in **Figure 1d, 2d** and **3c**.

“2. Regarding the mouse metabolic phenotyping, the authors should examine whole-body energy expenditure, food intake, and movement/activity of the mice under a regular diet as well. In addition, BAT thermogenesis capacity should be determined by measuring oxygen consumption rate, in addition to gene expression.”

RE: We have done these experiments and the data are shown in **Supplementary Fig 11a-e**.

“3. In Page6 line 111-112: the meaning of the sentence "traditional CHIP-seq analysis methods did not consider the global changes between samples" is unclear. Many studies have compared the data

between samples. The normalization methods in Fig.2A-B and the data without "normalization" should be provided and discuss the rationale."

RE: As shown in **Figure 1a** and **Supplementary Fig 2a**, our Western blot results indicate there are global differences in H3K36me2 and H3K27me3 levels between WT and K36M cells. However, these differences are not captured in ChIP-Seq analysis if one uses naïve normalization, where one simply normalizes the read counts by library size. Normalization only by library size essentially assumes that WT and K36M cells have similar histone modification levels. To address this issue, we used Western blot results for additional normalization. Specifically, each ChIP signal intensity was firstly normalized with each library size. Then, H3K36me2 or H3K27me3 ChIP signal intensity was further normalized with global H3K36me2 or H3K27me3 levels measured by Western blot. Details of this approach are now included in the Methods section.

In the figure below, we show the data without normalization by Western blot (the top panels). For comparison, we also show the data with normalization by western blot (the bottom panels, shown in **Supplementary Fig 2c-d**).

Data without normalization:

Supplementary Fig 2c-d:

Reviewer #2 (Remarks to the Author):

"This manuscript identifies a role for NSD2 and its catalytic product H3K36me2 in adipogenesis in vitro and for H3K36 methylation in vivo in WAT and BAT development and function. The authors show that in immortalized cell lines the expression of H3.3K36M or depletion of NSD2 (but not NSD1 or SETD2), both of which lead to loss of H3K36me2 and a concomitant increase in H3K27me3, block differentiation of immortalized preadipocytes into adipocytes. Genomic studies (RNA-seq and ChIP-seq) are performed in both K36M and NSD2-KD-expressing cells and support a model in which loss of H3K36me2 leads to re-distribution of H3K27me3 and repression of key developmental target genes. In vivo, the authors observe a compelling phenotype in mice expressing H3.3 K36M in adipocytes. While the observed phenotype is not entirely consistent with the one observed in their

tissue culture model (there does not appear to be a similar block in differentiation of pre-adipocytes), it is apparent that adipose tissue function is perturbed in the K36M mice, and there are changes in K36me2 and K27me3 that are consistent with the cell culture observations.”

RE: We thank Reviewer #2's careful reading of our manuscript and appreciate his/her highly constructive comments.

Our study aims to understand the role of histone H3K36 methylation in both development (adipogenesis) and function of adipose tissue. Since *Fabp4* (*aP2*) promoter-driven H3.3K36M expression is relatively late in adipogenesis^{2,3}, *aP2*-H3.3K36M transgenic mice (A-K36M) are suitable for studying the role of H3K36 in adipose tissue function but not adipogenesis. Indeed, *aP2* promoter-driven H3.3K36M expression has minimal effects on adipose tissue development (adipogenesis). Newborn A-K36M pups did not show significant defects in BAT development (**new Supplementary Fig 10a**). Adult A-K36M mice showed similar body weight and fat and lean mass, and similar weight and size of the BAT and WATs compared to those of Ctrl mice (**Figure 5d-f**).

To support the critical role of H3K36 methylation in adipogenesis *in vivo*, we have added data from a new transgenic mouse model (LSL-K36M, see the **new Figure 4**). We crossed LSL-K36M mice with *Myf5-Cre* mice to induce H3.3K36M expression in progenitor cells of BAT and muscle lineages^{4,5}. Consistent with *in vitro* adipogenesis data (**Figure 1**), expression of H3.3K36M in *Myf5*⁺ progenitor cells *in vivo* led to significant defects in BAT development (**new Figure 4**).

Regarding the role of H3K36 methylation in adipose tissue function, we previously showed that *aP2* promoter-driven H3.3K36M expression in adipocytes (A-K36M mice) reprograms gene expression and causes functional defects in adipose tissues (**Figure 5-7**). In the revised version, we show *in vitro* data consistent with the functional defects *in vivo*. In primary A-K36M brown adipocytes, *aP2* promoter-driven H3.3K36M expression changed H3K36/K27 methylation levels and increased white fat marker gene expression (**new Supplementary Fig 10**).

To help understanding of our findings, we have included a schematic in the **new Figure 8**. Reflecting our consistent data from *in vitro* and *in vivo* experiments, this model illustrates our conclusion that H3K36 methylation is important for adipose tissue development and function. In progenitor cells, depletion of H3K36 methylation by H3.3K36M leads to defects in adipose tissue development (adipogenesis). In mature adipocytes, depletion of H3K36 methylation by H3K36M reprograms functional gene expression, impairing brown and white adipose tissue functions.

“In summary, this manuscript identifies H3K36me2 (and thereby NSD2) crucial for normal fat development. The conclusion that this abundant histone modification is important for a developmental process is not surprising and the proposed mechanism (via H3K27me3 redistribution) has been previously described. Nonetheless, the thoroughness of the study, the *in vivo* mouse model work, and establishing the link between NSD2-H3K36me2 and proper physiologic fat development is significant and the work should be of broad interest to the field.”

RE: Regarding the reviewer's concern on novelties, we have updated the manuscript to include new data and to explicitly state the novelties of our study:

- a) By expressing H3K36M in progenitor cells of adipocytes, we show for the first time that H3K36 methylation is required for adipogenesis *in vivo* (**new Figure 4**).
- b) A recent Science paper reported that H3K36M **decreases** H3K27me3 levels on mesenchymal stem cell (MSC) genes, which allows MSC gene expression to inhibit cell differentiation including osteogenesis and adipogenesis¹. Using preadipocyte differentiation as the model system, we show two novel mechanisms by which H3K36M inhibits adipogenesis. First, H3K36M **increases** H3K27me3 levels on the gene locus encoding a master adipogenic transcription factor C/EBP α , which prevents C/EBP α expression (**Figure 1g**). Second, H3K36M targets *Nsd2* to inhibit ligand-induced expression of target genes of PPAR γ , the other master regulator of adipogenesis (**Figure 2**).

c) By expressing H3K36M in adipocytes, we show for the first time that H3K36 methylation is important for adipose tissue function in vivo (**Figure 5 – Figure 7**).

“Comments:

1) Figures 1 & 2: While the results appear similar, no comparisons are made between changes in gene expression and K36/K27 methylation. The data should be analyzed more extensively. For example, in Fig. 1D, a distinction is made between “K36M-sensitive” and “K36M-resistant” DEGs upon differentiation. The authors seem to suggest that these changes in the K36M category are due to changes in K36/K27 methylation, but this is only explored at 4 hand-selected genes in Fig 2E. To make better use of the data they already have, the authors should generate metaplots similar to Fig 2C and 2D, but instead of “All genes”, looking at “K36M-resistant genes” and “K36M-sensitive genes”. The authors’ model would predict that changes in K36/K27 might be observed in the latter and not the former.”

RE: We have followed the reviewer’s suggestion and generated metaplots for the comparison of H3K36me2 and H3K27me3 levels between K36M-sensitive and -resistant genes (**new Supplementary Fig 3**). This analysis suggests that H3.3K36M represses the induction of K36M-sensitive genes through decreased H3K36me2 and increased H3K27me3 levels on these genes.

“2) Can the authors comment on the distribution of H3.3 K36M? Is it enriched at the affected loci, directly inhibiting K36me2 in cis, or are the observed effects predominantly a trans-effect? ChIP on the FLAG-tagged H3.3 at the four loci in Fig. 2E would be informative.”

RE: We did not perform ChIP of the FLAG-tagged H3.3 because FLAG antibody never worked for ChIP in our hands. We show in **Supplementary Fig 1a-c** that H3.1K36M and H3.3K36M have similar effects on H3K36me2 levels, adipogenesis, and adipogenic gene expression. These results are consistent with a previous study¹, which reported different distributions of H3.1K36M and H3.3K36M but similarly altered gene expression profiles between H3.1K36M- and H3.3K36M-expressing mesenchymal stem cells (MSCs). Together, these results suggest a substantial *trans*-effect of H3K36M. We cannot rule out the potential involvement of a *cis*-effect of H3K36M. It has been shown previously that H3.3K36M incorporated in nucleosomes reduced H3K36me2 and H3K36me3 levels locally⁹, suggesting a *cis*-effect. However, since the expression levels of H3.3K36M were much lower than that of endogenous H3 (**Figure 1a**), the *cis*-effect of H3K36M on chromatin is unlikely to play a major role in the cells used in this study.

“3) Figure 4: Similar to Fig. 1 & 2, the information presented in Fig. 4 appears to be consistent with results observed in K36M cells, but the direct comparison is never made. Specifically, Fig. 4D is intentionally displayed in a similar format to Fig. 1D; but do the categories defined in each figure overlap? A Venn diagram comparing the categories defined in Fig. 1D and Fig. 4D should be included in Figure 4.”

RE: We have followed the reviewer’s suggestion and added a Venn diagram in the **new Figure 3e** to compare Nsd2-dependent (**current Figure 3c**) and K36M-sensitive (**Figure 1d**) gene groups. We have also done gene set enrichment analysis (GSEA) to directly compare genes affected by H3.3K36M expression and Nsd2 KD (**new Supplementary Fig 7**). These results provide additional supports for the conclusion that H3.3K36M targets Nsd2 to inhibit adipogenesis.

“4) Knock-down of NSD2 is known to be toxic to many cell types and can cause loss of proliferation. Thus, are the effects on adipogenesis a direct result of the loss of H3K36me2 due to expression of H3.3K36M and NSD2 KD or is it a secondary effect of toxicity associated with depletion of H3K36me2? To rule out the possibility of toxicity, the authors should determine growth rates of cells expressing H3.3K36M and control as well as Control, shNSD1, shNSD2, and shSETD2 cells without differentiation.”

RE: In HT1080 fibrosarcoma cells, H3.3K36M expression or Nsd2 KD leads to decreased cell proliferation¹⁰. However, in mesenchymal progenitor cells, H3.3K36M causes cell proliferation rates

higher than H3.3WT^{1,9}. We have followed the reviewer's suggestion and examined the growth rates of immortalized preadipocytes expressing H3.3K36M and H3.3WT as well as cells with *Nsd1* KD, *Nsd2* KD and *Setd2* KD (**new Supplementary Fig 6a**). *Nsd1* or *Setd2* KD did not affect cell growth rates. Although H3.3K36M expression or *Nsd2* KD slightly decreased the growth rate of immortalized brown preadipocytes, these cells could grow to be over-confluent for inducing differentiation, indicating that the adipogenesis defects observed in these cells are not mainly due to the slight growth defects.

"5) It is surprising that NSD1 KD led to a detectable loss of H3K36me2 (Fig S3) as the majority of H3K36me2 being generated in these cells appears to be due to NSD2 (e.g. Fig 4). Please comment."
RE: We have done ChIP analyses to compare *Nsd1* and *Nsd2* KD on H3K36me2 levels around selected adipogenic gene loci. As shown in the **new Supplementary Fig 6b**, *Nsd2* KD has more significant effects on H3K36me2 levels around gene loci of *Cebpa*, *Lpl* and *Cd36*, suggesting that *Nsd2*-mediated H3K36me2 plays a more important role in regulating adipogenic gene expression.

References:

1. Lu, C. *et al.* Histone H3K36 mutations promote sarcomagenesis through altered histone methylation landscape. *Science (New York, N.Y.)* **352**, 844-849 (2016).
2. Kang, S., Kong, X.X. & Rosen, E.D. Adipocyte-Specific Transgenic and Knockout Models. *Method Enzymol* **537**, 1-16 (2014).
3. Wang, L. *et al.* Histone H3K9 methyltransferase G9a represses PPAR[gamma] expression and adipogenesis. *EMBO J* **32**, 45-59 (2013).
4. Lee, J.E. *et al.* H3K4 mono- and di-methyltransferase MLL4 is required for enhancer activation during cell differentiation. *eLife* **2**, e01503 (2013).
5. Seale, P. *et al.* PRDM16 controls a brown fat/skeletal muscle switch. *Nature* **454**, 961 (2008).
6. Reue, K. & Phan, J. Metabolic consequences of lipodystrophy in mouse models. *Curr Opin Clin Nutr* **9**, 436-441 (2006).
7. Kosmiski, L.A. *et al.* Total energy expenditure and carbohydrate oxidation are increased in the human immunodeficiency virus lipodystrophy syndrome. *Metabolism* **52**, 620-625 (2003).
8. He, W. *et al.* Adipose-specific peroxisome proliferator-activated receptor gamma knockout causes insulin resistance in fat and liver but not in muscle. *Proc Natl Acad Sci U S A* **100**, 15712-15717 (2003).
9. Fang, D. *et al.* The histone H3.3K36M mutation reprograms the epigenome of chondroblastomas. *Science (New York, N.Y.)* **352**, 1344-1348 (2016).
10. Sankaran, S.M. & Gozani, O. Characterization of H3.3K36M as a tool to study H3K36 methylation in cancer cells. *Epigenetics*, 0 (2017).

Reviewers' comments:

Reviewer #1 (Remarks to the Author):

The revised manuscript by Zhuang et al. provides substantial amounts of new data that support the authors' conclusions. The authors also addressed most of the reviewer's comments well. Notably, new data on a mouse model that ectopically expresses H3.3 K36M in Myf-5 lineage support a critical role of H3.3K36 during the early phase of adipocyte differentiation.

Regarding the new data in Fig. 4, the authors should investigate whether changes in H3.3K36M affect cell fate determination of brown adipocytes vs. skeletal muscle, or if general differentiation program is impaired both in Myf5-lineage brown adipocytes and myocytes in mouse embryos. The result would influence the conclusion model in Fig. 8.

Regarding the original comment #1, the authors should examine if any change in cell death (apoptosis or necrosis) is seen in the adipose tissue (WAT) of aP2-K36K transgenic mice. Given the striking difference in adipose tissue mass between control and Tg mice, it is possible that additional mechanisms may contribute to the difference in fat mass in addition to reduced lipolysis and impaired expansion of adipocytes.

Reviewer #2 (Remarks to the Author):

The authors have addressed all of the major issues in their revised manuscript and I recommend publication of the study.

Point-by-point responses:

Reviewer #1 (Remarks to the Author):

“The revised manuscript by Zhuang et al. provides substantial amounts of new data that support the authors' conclusions. The authors also addressed most of the reviewer's comments well. Notably, new data on a mouse model that ectopically expresses H3.3 K36M in Myf-5 lineage support a critical role of H3.3K36M during the early phase of adipocyte differentiation.

Regarding the new data in Fig. 4, the authors should investigate whether changes in H3.3K36M affect cell fate determination of brown adipocytes vs. skeletal muscle, or if general differentiation program is impaired both in Myf5-lineage brown adipocytes and myocytes in mouse embryos. The result would influence the conclusion model in Fig. 8.”

RE: We thank Reviewer #1 for careful reading of our updated manuscript and appreciate the highly constructive comments. We have several lines of evidence to indicate that the general differentiation program is impaired by H3.3K36M. 1) Expression of H3.3K36M in C2C12 cells impairs myogenesis (Supplementary Fig 1d-f). 2) Expression of H3.3K36M in Myf5⁺ progenitor cells leads to reduction of both BAT and muscle mass in mice (Fig 4b-d). 3) We have added **new Fig 4h-j** to show that in LSL-K36M preadipocytes, induction of H3.3K36M expression inhibits MyoD-driven myogenesis. Together, these results indicate that H3K36M impairs general differentiation program both in Myf5-lineage brown adipocytes and myocytes in mouse embryos. Consistently, a previous study (Lu, Jain et al. 2016) reported that expression of H3K36M in mesenchymal stem cells inhibited cell differentiation including osteogenesis, chondrogenesis and adipogenesis. We have updated the title of Fig. 4 and our model in Fig. 8.

“Regarding the original comment #1, the authors should examine if any change in cell death (apoptosis or necrosis) is seen in the adipose tissue (WAT) of aP2-K36M transgenic mice. Given the striking difference in adipose tissue mass between control and Tg mice, it is possible that additional mechanisms may contribute to the difference in fat mass in addition to reduced lipolysis and impaired expansion of adipocytes.”

RE: We'd like to point out that the striking difference in adipose tissue mass between control (Ctrl) and A-K36M (aP2-K36M) mice is only found when mice are fed with high fat diet (HFD), but not with regular diet (Fig 5 and 6). We have updated Fig 7c to include the WAT weight of mice before HFD. Together with Supplemental Fig 12a, these data indicate a moderate increase of fat mass in A-K36M mice after HFD.

We have followed the reviewer's suggestion and checked cell death in WATs after HFD using TUNEL assay. As shown in the figure below, no positive TUNEL signal was detected in epi-WAT or ing-WAT of Ctrl and A-K36M mice after HFD (left panels). A positive control from another experiment performed by the same technician confirmed that our TUNEL kit worked well (the right panel). Thus, although we can't rule out the contribution of additional mechanisms, cell death is unlikely one of them.

Reviewer #2 (Remarks to the Author):

"The authors have addressed all of the major issues in their revised manuscript and I recommend publication of the study."

RE: We greatly appreciate Reviewer #2's favorable review of our updated manuscript.

Reference:

Lu, C., et al. (2016). "Histone H3K36 mutations promote sarcomagenesis through altered histone methylation landscape." *Science* **352**(6287): 844-849.

REVIEWERS' COMMENTS:

Reviewer #1 (Remarks to the Author):

The authors have addressed the reviewer's comments satisfactory.